# Variations in Nitrogen Accumulation and Use Efficiency in Maize Differentiate with Nitrogen and Phosphorus Rates and Contrasting Fertilizer Placement Methodologies

**DOI:** 10.3390/plants12223870

**Published:** 2023-11-16

**Authors:** Sharifullah Sharifi, Songmei Shi, Xingshui Dong, Hikmatullah Obaid, Xinhua He, Xirong Gu

**Affiliations:** 1National Base of International S&T Collaboration on Water Environmental Monitoring and Simulation in the Three Gorges Reservoir Region, Chongqing Key Laboratory of Plant Resource Conservation and Germplasm Innovation, School of Life Sciences, College of Resources and Environment, Southwest University, Chongqing 400715, China; nsharifullah@gmail.com (S.S.); dxshui@email.swu.edu.cn (X.D.); hikmat_obaid@yahoo.com (H.O.); 2Department of Soil Science and Irrigation Management, Faculty of Plant Sciences, Afghanistan National Agricultural Sciences and Technology University (ANASTU), Kandahar 3801, Afghanistan; 3School of Horticulture and Landscape, Yunnan Agricultural University, Kunming 650201, China; shismei@email.swu.edu.cn; 4School of Biological Sciences, University of Western Australia, Perth 6009, Australia; 5Department of Land, Air and Water Resources, University of California at Davis, Davis, CA 90616, USA

**Keywords:** broadcast, deep band, nitrogen accumulation, nitrogen use efficiency, side band, *Zea mays*

## Abstract

Balanced nitrogen (N) and phosphorus (P) rates, coupled with rational fertilization methodology, could promote crop N accumulation, N use efficiency, and yield production, particularly in semi-arid and arid regions. To test these characteristics, a two-year (2018 and 2019) pot experiment was performed by growing summer maize in a rain-proof glass greenhouse under nine combined N (112, 150, and 187 kg ha^−1^, urea) and P (45, 60, and 75 kg ha^−1^ calcium superphosphate) rates and three contrasting fertilizer placements. The fertilizers were placed by broadcast on the soil surface (Broadcast), a side band on a 4 cm strip of soil surface within 7 cm from the sowing line (Side band), and a deep band on a 4 cm strip below 7 cm soil depth within 7 cm from the sowing line (Deep band). Results from three maize growth stages (eight-leaf, 45 days after sowing, DAS; tasseling, 60 DAS; and harvest, 115 DAS) showed that leaf, stem, root N accumulation, and total soil N were significantly increased under Deep band than under both Side band and Broadcast at N150P60, N187P60, N150P75, and N187P75, but not at N112P45, N150P45, N187P45, N112P60, and N112P75. Significantly greater leaf, stem, and root N accumulations were also displayed at N150 and N187 than at N112 for the same P60 or P75 under the Deep band at 60 DAS and 115 DAS; while for leaf and stem, N accumulations were greater at P75 and P60 than at P45 for the same N150 under Deep band at 45 DAS, 60 DAS, and 115 DAS. Significantly greater agronomy N use efficiency, partial factor productivity, and N use efficiency were exhibited under the Deep band than under the Side band and Broadcast at N150P75 and N187P75, but at N150P60 and N187P60 for NUE only. In addition, leaf, stem, seed, and root N concentrations positively correlated with their own N accumulations or soil N concentrations at the tasseling and harvest stages. Our results demonstrate that a synchronized N150P60, N187P60, N150P75, or N187P75 fertilization rate with Deep band placement can improve soil N availability and root N uptake, and thereby, increase aboveground N accumulation, N use efficiency, and yield production of maize, which is particularly practical for small-holder farmers globally.

## 1. Introduction

As the largest soil nutrient required by plants, nitrogen (N, either ammonium (NH_4_^+^) or nitrate (NO_3_^−^), is an essential element for the production of chlorophylls, amino acids, proteins, nucleic acids, phytohormones, and secondary metabolites [1,2]. To meet the food supply for an ever-increasing population, nitrogenous fertilizers are increasingly applied to enhance crop production from 11.5 million tons in 1961 to 108.7 million tons in 2021 worldwide [3]. However, ~46–67% of N fertilizers applied to soil are lost mainly through NH_4_^+^ volatilization, NO_3_^−^ leaching, nitrification-denitrification, and surface runoff, resulting in low N use efficiency (NUE) and crop yield, less economic return, air pollution, and waterbody eutrophication, thus the risk of sustainable agriculture [4,5]. Best fertilization practices can mitigate these losses while enhancing crop productivity, profitability, and nutrient use efficiency [6]. 

Maize (*Zea mays* L.) ranks as the third staple crop after wheat and rice, with an annual production of 0.27 million tons from 0.14 million hectares of plantations in Afghanistan [7,8], where most soil characteristics are a high pH, low organic matter and moisture, and low N and P availability under a semi-arid and arid climate [9,10]. Mainly due to a lack of chemical fertilizer supply, plus a high price, and poor field management, the averaged maize yield is quite low, at 1.93–2.20 t ha^−1^ in Afghanistan compared to 5.88 t ha^−1^ globally [7,8,9,10]. A total of 0.043 million tons of N fertilizers (mostly urea, ammonium sulfate, and diammonium phosphate) were applied to increase crop productivity in Afghanistan in 2020 [3]. For maize production in Afghanistan, an averaged 90 kg N/33 kg P ha^−1^ (N90P33) fertilization rate is widely adopted [11], while new studies have recommended an averaged N160P70 as an optimum fertilization rate [9,12,13,14]. 

Methodologies of applying chemical N and/or P fertilizers for field crops are broadly categorized into surface broadcast and band placement or fertilization [15,16]. Obviously, through fertilizers being spread on the soil surface, broadcast is the most popular, fastest, yet simplest labor- and time-saving, fertilization methodology. A side band or deep band placement, which is to place fertilizer close to and alongside the sowing seeds/emerging roots and/or in a deeper soil depth, is more effective in warranting a high nutrient availability by increasing root-fertilizer contact/uptake while improving NUE, mitigating nutrient losses, and greenhouse gas emissions [6,15,16,17,18]. Numerous studies on a number of crops, including barley, maize, oats, potatoes, rapeseed, rye, rice, sorghum, soybeans, wheat, etc., have shown such diverse benefits [15,16]. For instance, compared to urea broadcast or at 8 cm depth, the Deep band at 20 cm depth generated higher silage or grain yields of maize and thus the highest N agronomic efficiency (NAE) [19]. Both 5 or 10 cm deep fertilizer placements aside from 5 cm of the maize seeds decreased the fertilizer costs and the risk of N losses to ground and surface waters [20]. Compared to the surface broadcast, the application of urea in the maize root zone (5 cm from seeds at 12 cm soil depth) effectively fulfilled N demand while reducing 20–25% N losses [21]. However, there could be fewer nutrients available to fewer roots as the distance of fertilizer from the root zone increases [20], while high amounts of applied nutrients would probably be toxic to adjoining germinating seeds and/or emerging roots [22]. Hence, fertilization at a reasonably vertical soil depth to root elongation and a horizontal distance for the seeding zone paves the way for greater nutrient supplement and root uptakes. Nevertheless, at present, the most common fertilizer placement is broadcast, whereas Deep band placement is also applied, while few studies have compared their advantages in improving fertilizer use efficiency and crop production. As a result, a rational N/P fertilization rate and practical fertilizer placement methodology are needed to promote crop production.

By simulating the current fertilization practices (N/P fertilizer combination and rates as well as fertilizer placement methodologies) in maize plantations in semi-arid and arid Afghanistan, the present study employed three contrasting fertilizer placements (Broadcast, Side band, and Deep band, see their details in the Materials and Methods section) to address which fertilizer placement and fertilization rate could be most desirable to increase N accumulations in both soil and maize tissues, and hence a better NUE and yield production. In doing so, the objectives of this study were to identify (1) an optimum NP fertilization rate; (2) a practical fertilizer placement methodology; and (3) a positive correlation between an appropriate NP rate with proper placement and tissue N accumulations for increasing maize production. The generated results will be helpful in contributing to field practices in exploring effective fertilization rates and methodologies for small-holder farmers to increase crop production around the world. 

## 2. Results

### 2.1. Leaf N Concentration and Accumulation 

#### 2.1.1. Leaf N Concentration 

Leaf N concentration (g kg^−1^) was generally highest under the no-fertilization control at 45 DAS, followed by at 60 DAS and lowest at 115 DAS (Figure 1A,E,I), irrespective of the NP fertilization rate and fertilizer placement. There were significant differences in leaf N concentration among different growth days for the same N and P fertilization rates (N112P45, N112P60, N112P75, N150P45, N150P60, N150P75, N187P45, N187P60, or N187P75) and same fertilizer placement (Broadcast, Side band, or Deep band) in the order of 45 DAS < 60 DAS ≤ 115 DAS (Figure 1B–D,F–H,J–L).

Leaf N concentration was similar among three different fertilizer placements of Broadcast, Side band, and Deep band with the same N and P fertilization rates under N112P45, N150P45, and N187P45 at 45 DAS, 60 DAS, or 115 DAS (Figure 1B–D), under N112P60, N150P60, and N187P60 at 45 DAS or 115 DAS, under N112P60 at 60 DAS (Figure 1F–H), under N112P75 at 45 DAS, 60 DAS or 115 DAS, under N150P75 at 45 DAS, also under N187P75 at 45 DAS and 115 DAS (Figure 1J–L). In addition, leaf N concentration was significantly greater in the Deep band than in Broadcast, and then in Side band under N150P60, N187P60, and N187P75 at 60 DAS (Figure 1G,H,L), under N150P75 at 60 DAS and 115 DAS (Figure 1K).

Leaf N concentration was similar among different N fertilization rates of N112, N150, and N187 with the same P fertilization rates either at P45 or P60 and P75 and the same fertilizer placements under Deep band, Side band, and Broadcast at 45 DAS, 60 DAS, or 115 DAS (Figure 1B–D,F–H,J–L). Leaf N concentration was similar among different P fertilization rates of P45, P60, and P75 for the same N112 (Figure 1B,F,J), N150 (Figure 1C,G,K), and N187 (Figure 1D,H,L) under Deep band, Side band, and Broadcast at 45 DAS, 60 DAS or 115 DAS. Leaf N concentration was significantly greater in P75 than in P45 for N150 under the Deep band at 60 DAS and 115 DAS, as well as for N187 under the Deep band at 60 DAS (Figure 1C,D,G,H,K,L).

#### 2.1.2. Leaf N Accumulation

Significantly leaf N accumulation (mg plant^−1^) among growth days in the no-fertilization control was ranked as 115 DSA > 60 DAS > 45 DAS (Figure 2A,E,I), irrespective of NP fertilization rate and fertilizer placement.

Leaf N accumulation varied significantly among different growth days for the same N and P fertilization rate (N112P45, N112P60, N112P75, N150P45, N150P60, N150P75, N187P45, N187P60, or N187P75) and same fertilizer placements (Broadcast, Side band, and Deep band). Leaf N accumulation was significantly greater at 60 DAS and 115 DAS than at 45 DAS (Figure 2B–D,F–H,J–L). Among different fertilizer placements for the same N and P fertilization rate, leaf N accumulation was similar between Broadcast, Side band, and Deep band under N112P45 and N150P45 at 60 DAS and 115 DAS, under N187P45 at 45 DAS, 60 DAS or 115 DAS (Figure 2B–D), under N112P60 at 60 DAS and 115 DAS (Figure 2F), also under N112P75 at 60 DAS and 115 DAS (Figure 2J). Meanwhile, the Deep fertilizer placement significantly increased leaf N accumulation than the Side band and Broadcast under N150P60, N187P60, N150P75, and N187P75 at 45 DAS, 60 DAS, and 115 DAS (Figure 2G,H,K,L). In addition, leaf N accumulation was significantly enhanced in the Deep band than in Broadcast under N112P45, N150P45, N112P60, and N112P75 at 45 DAS (Figure 2B,C,F,J). No significant differences were observed in leaf N accumulation among different N fertilization rates of N112, N150, and N187 with the same P45 fertilization rate under Broadcast, Side band, and Deep band at 45 DAS, 60 DAS, or 115 DAS (Figure 2B–D). Leaf N accumulation among different N fertilization rates with the same P60 or P75 fertilization rates was significantly greater in N150 and N187 than in N112 under the Deep band at 45 DAS, 60 DAS, or 115 DAS (Figure 2F–H,J–K). Among different P fertilization rates with the same N fertilization rate and same fertilizer placement, the P75 significantly increased leaf N accumulation than P60 and P45 for N150 under Broadcast at 45 DAS, for N187 under the Deep band at 45 DAS (Figure 2C,D,G,H,K,L). Similarly, leaf N accumulation was significantly greater in P75 and P60 than in P45 for N150 under the Deep band at 45 DAS, 60 DAS, or 115 DAS (Figure 2C,G,K). Moreover, leaf N accumulation was significantly greater in P75 than in P45 for N112 and N187 under Broadcast at 45 DAS (Figure 2B,D,F,H,J,L), for N112, N150, and N187 under the Side band at 45 DAS (Figure 2B–D,F–H,J–L), also for N187 under Deep band at 60 DAS (Figure 2D,H,L). Meanwhile, leaf N accumulation was similar among different P fertilization rates of P75, P60, and P45 for the same N112, N150, and N187 fertilization rates under the Side band and Broadcast at 60 DAS and 115 DAS, for N112 under the Deep band at 45 DAS, 60 DAS, or 115 DAS, and for N187 under Deep band at 115 DAS (Figure 2B–D,F–H,J–L).

### 2.2. Stem N Concentration and Accumulation

#### 2.2.1. Stem N Concentration 

In the no-fertilization control treatment, significantly greater stem N concentration (g kg^−1^) was in the order of 45 DAS > 60 DAS > 115 DAS (Figure 3A,E,I), irrespective of NP fertilization rate and fertilizer placement.

There were significant differences in stem N concentration among different growth days for the same N and P fertilization rate and the same fertilizer placement observed at 45 DAS than at 60 DAS and 115 DAS for N112P45 under Broadcast, Side band, and Deep band (Figure 3B), for N187P45, N150P60, and N187P75 under Broadcast (Figure 3D,G,L), for N11260 and N11275 under Broadcast and Side band (Figure 3F,J), and for N150P45 under Broadcast and Deep band (Figure 3C). Moreover, significant differences in stem N concentration were in the order of 45 DAS < 60 DAS < 115 DAS for N187P45, N150P60, and N187P75 under the Side band and Deep band (Figure 3D,G,L), for N112P60 and N112P75 under the Deep band (Figure 3F,J), for N187P60 and N150P75 under Broadcast, Side band and Deep band (Figure 3H,K), as well as N150P45 under the Side band (Figure 3C).

Stem N concentration was similar among three different fertilizer placements (Deep band, Side band, and Broadcast) for the same N and P fertilization rates under N112P45, N150P45, N187P45, N112P60, N150P60, and N112P75 at 45 DAS, 60 DAS, or 115 DAS (Figure 3B–D,F–G,J), under N187P60 at 45 DAS or 115 DAS (Figure 3H), under N150P75 and N187P75 at 115 DAS (Figure 3K,L). Meanwhile, a significantly greater stem N concentration among different fertilizer placements for the same N and P fertilization rate was observed in the Deep band than in the Side band and Broadcast under N150P75 and N187P75 at 45 DAS and 60 DAS, and under N187P60 at 60 DAS (Figure 3H,K,L).

Stem N concentration was similar among different N fertilization rates of N112, N150, and N187 for the same P fertilization rates either at P45 or P60 and P75 and the same fertilizer placement under Deep band, Side band, and Broadcast at 45 DAS, 60 DAS, or 115 DAS (Figure 3B–D,F–H,J–L).

Stem N concentration was similar among different P fertilization rates of P45, P60, and P75 for the same N112 fertilization rates under Broadcast, Side band and Deep band (Figure 3B,F,J), for the same N150 (Figure 3C,G,K) and N187 fertilization rates (Figure 3D,H,L) under Side band and Broadcast at 45 DAS, 60 DAS, or 115 DAS, and for N150P75 under Deep band at 115 DAS (Figure 3C,G,K). In addition, stem N concentration among different P fertilizations for the same N fertilization rate and same fertilizer placement was significantly greater in P75 and P60 than in P45 for N150P75 under Deep band at 60 DAS (Figure 3C,G,K), and for N187P75 under the Deep band at 45 DAS and 60 DAS (Figure 3D,H,L). Similarly, a significantly greater stem N concentration was observed in P75 than in P45 for N150 (Figure 3C,G,K) under the Deep band at 45 DAS, as well as N187 under the Deep band at 115 DAS (Figure 3C,G,K).

#### 2.2.2. Stem N Accumulation

Stem N accumulation (mg plant^−1^) in the no-fertilization control treatment was ranked as 115 DAS > 60 DAS > 45 DAS (Figure 4A,E,I), irrespective of NP fertilization rate and fertilizer placement.

There were significant differences in stem N accumulation among different growth days for the same N and P fertilization rate (N112P45, N112P60, N112P75, N150P45, N150P60, N150P75, N187P45, N187P60, or N187P75) and same fertilizer placement (Broadcast, Side band, and Deep band). They were ranked as 115 DAS ≈ 60 DAS > 45 DAS (Figure 4B–D,F–H,J–L). A significantly greater N accumulation was observed at 115 DAS > 60 DAS > 45 DAS for N150P60 and N187P60 with Side band (Figure 4G,H). Stem N accumulation among different fertilizer placements for the same N and P fertilization rate was significantly greater in the Deep band than in the Side band and Broadcast under N150P60, N150P75, N187P60, and N187P75 at 45 DAS, 60 DAS, or 115 DAS (Figure 4G,H,K,L). Similarly, a significantly greater stem N accumulation was observed in the Deep band than in Broadcast under N112P45 at 45 DAS and 115 DAS, and under N150P45 and N187P45 at 45 DAS (Figure 4B–D). In addition, stem N accumulation was not significantly different among the Deep band, Side band, and Broadcast under N150P45 and N187P45 at 60 DAS and 115 DAS (Figure 4C,D), under N112P45 at 45 DAS (Figure 4B), and under N112P60 and N112P75 at 45 DAS, 60 DAS or 115 DAS (Figure 4F,J).

Stem N accumulation was similar among the different N fertilization rates of N112, N150, and N187 with the same P45 fertilization rate under Deep band, Side band, and Broadcast at 45 DAS, 60 DAS, or 115 DAS (Figure 4B–D), except for a significantly greater stem N accumulation in N187 than in N112 under Broadcast at 45 DAS (Figure 4B,D). Stem N accumulation among different N fertilization rates with the same P60 or P75 fertilization rate was also significantly greater in N187 and N150 than in N112 under Deep band at 45 DAS, 60 DAS, or 115 DAS, while similar stem N accumulation among different N fertilization rates with the same P60 or P75 fertilization was observed in N112, N150 and N187 under Side band at 45 DAS, 60 DAS or 115 DAS, and under Broadcast at 60 DAS and 115 DAS (Figure 4F–H,J–L).

Stem N accumulation was similar among the different P fertilization rates of P45, P60, and P75 for the same N112 fertilization rate under Deep band, Side band, and Broadcast at 45 DAS, 60 DAS, or 115 DAS (Figure 4B,F,J), for N150 under Side band and Broadcast at 45 DAS, 60 DAS, or 115 DAS (Figure 4C,G,K), as well as N187 under Broadcast at 60 DAS and 115 DAS, and under Side band at 45 DAS, 60 DAS, or 115 DAS, and under Deep band at 115 DAS (Figure 4D,H,L). Moreover, stem N accumulation was significantly greater in P75 and P60 than in P45 for N150 under the Deep band at 45 DAS and 60 DAS (Figure 4C,G,K). In addition, stem N accumulation was significantly greater in P75 than in P45 for N187 under Broadcast and Deep band at 45 DAS, under Deep band at 60 DAS (Figure 4D,H,L), and for N150 under Deep band at 115 DAS (Figure 4C,G,K).

### 2.3. Root N Concentration and Accumulation

#### 2.3.1. Root N Concentration

In the no-fertilization control, root N concentration (g kg^−1^) was similar among 60 DAS and 115 DAS, and it decreased with plant aging (Figure 5A,E,I), regardless of NP fertilization rate and fertilizer placement.

Root N concentration was similar among different growth days at 60 DAS and 115 DAS with the same N and P fertilization rates (N112P45, N112P60, N112P75, N150P45, N150P60, N150P75, N187P45, N187P60, or N187P75) and same fertilizer placements (Broadcast, Side band, and Deep band) (Figure 5B–D,F–H,J–L), except for a significantly lower root N concentration at 115 DAS than at 60 DAS for N187P60 under Broadcast (Figure 5H).

Root N concentration was similar among three different fertilizer placements (Broadcast, Side band, and Deep band) for the same N and P fertilization rates (N112P45, N112P60, N112P75, N150P45, N150P60, N150P75, N187P45, N187P60, or N187P75) at 60 DAS and 115 DAS (Figure 5B–D,F–H,J–L).

No significant difference was observed in root N concentration among different N fertilization rates of N112, N150, and N187 for the same P45 (Figure 5B–D), or P60 (Figure 5F–H) and P75 (Figure 5J–L) fertilization rates under Deep band, Side band, and Broadcast at 60 DAS and 115 DAS, except for a significantly greater root N concentration in N187P60 than in N150P60 and N112P60 under Broadcast at 60 DAS (Figure 5F–H).

Root N concentration was similar among different P fertilization rates of P45, P60, and P75 for the same N112 (Figure 5B,F,J), N150 (Figure 5C,G,K), and N187 (Figure 5D,H,L) fertilization rates under Deep band, Side band, and Broadcast at 60 DAS and 115 DAS, except for a significantly greater root N concentration in P75 than in P45 for N112 under Broadcast at 115 DAS (Figure 5B,F,J). 

#### 2.3.2. Root N Accumulation

Root N accumulation (mg plant^−1^) in the no-fertilization control treatment was significantly greater at 115 DAS than at 60 DAS (Figure 6A,E,I), irrespective of NP fertilization rate and fertilizer placement.

Root N accumulation among different growth days for the same N and P fertilization rate and same fertilizer placement was significantly greater at 115 DAS than at 60 DAS for N150P45, N187P45, N112P60, N150P60, N187P60, and N187P75 under Deep band, Side band, and Broadcast (Figure 6C,D,F–H,L), for N112P45 and N112P75 under Side band and Broadcast (Figure 6B,J), as well as N150P75 under Side band (Figure 6K). In addition, root N accumulation was similar among 60 DAS and 115 DAS for N112P45 and N112P75 under Broadcast (Figure 6B,J), and N150P75 under Deep band and Broadcast (Figure 6K).

Among different fertilizer placements for the same N and P fertilization rate, the Deep band significantly increased root N accumulation than the Side band and Broadcast under N150P60, N187P60, N150P75, and N187P75 at 60 DAS and 115 DAS (Figure 6G,H,K,L), Similarly, root N accumulation was significantly greater in the Deep band than in Broadcast under N150P45 at 60 DAS, and under N112P45 at 115 DAS (Figure 6B,C). In addition, no significant differences in root N accumulation were observed among the Deep band, Side band, and Broadcast under N112P45 at 60 DAS (Figure 6B), under N150P45 at 115 DAS (Figure 6C), under N187P45 at 60 DAS and 115 DAS (Figure 6D), as well as under N112P60 and N112P75 at 60 DAS and 115 DAS (Figure 6F,J).

Root N accumulation among different N fertilization rates for the same P45 fertilization rate was significantly greater in N187 than in N112 under the Deep band at 60 DAS and 115 DAS, and under the Broadcast and Side band at 115 DAS (Figure 6B,D), while root N accumulation was similar between N112, N150, and N187 under Broadcast and Side band at 60 DAS and 115 DAS (Figure 6B–D). There was significantly greater root N accumulation among different N fertilization rates for the same P60 fertilization rate in N187 and N150 than in N112 under the Deep band at 60 DAS and 115 DAS (Figure 6F–H). Root N accumulation was significantly greater in N187 than in N112 under Broadcast at 60 DAS and 115 DAS, and under Side band at 115 DAS (Figure 6F,H), while similar root N accumulation was found between N112, N150, and N187 under Side band at 60 DAS (Figure 6F–H). In addition, root N accumulation among different N fertilization rates with the same P75 fertilization rate was significantly greater in N150 and N187 than in N112 under the Deep band at 60 DAS and 115 DAS (Figure 6J–L). Root N accumulation was also significantly greater in N187 than in N112 under the Side band at 115 DAS (Figure 6J,L), while root N accumulation was found similar among N112, N150, and N187 under Broadcast at 60 DAS, and 115 DAS and under Side band at 60 DAS (Figure 6J–L).

Root N accumulation among different P fertilization rates with the same N fertilization rate and same fertilizer placement was significantly greater in P75 and P60 than in P45 for N112 under the Side band at 115 DAS (Figure 6B,F,G), for N150 under Broadcast and Deep band at 115 DAS (Figure 6C,G,K), as well as for N187 under Deep band at 60 DAS and 115 DAS (Figure 6D,H,L). Likewise, root N accumulation was significantly greater in P75 than in P45 for N112 under Broadcast at 115 DAS (Figure 6B,F,J), and for N150 under Deep band at 60 DAS (Figure 6C,G,K). However, there was no significant difference in root N accumulation between P45, P60, and P75 for N112 under the Broadcast and Side band at 60 DAS and under the Deep band at 60 DAS and 115 DAS (Figure 6B,F,J), for N150 under Broadcast at 60 DAS and under the Side band at 60 DAS and 115 DAS (Figure 6C,G,K), also for N187 under Broadcast and Side band at 60 DAS and 115 DAS (Figure 6D,H,L).

### 2.4. Seed N Concentration and Accumulation

#### 2.4.1. Seed N Concentration

Seed N concentration (g kg^−1^) in the no-fertilization control was similar among different fertilizer placements under the Deep band, Side band, and Broadcast at 115 DAS (Figure 7A), irrespective of NP fertilization rate.

No significant difference was observed in seed N concentration among different fertilizer placements (Deep band, Side band, and Broadcast) for the same N and P fertilization rates (N112P45, N150P45, N187P45, N112P60, N150P60, N187P60, N112P75, N150P75, and N187P75) at 115 DAS (Figure 7B–D). Meanwhile, seed N concentration was similar among different N fertilization rates of N112, N150, and N187 for the same P fertilization rates either at P45, P60, or P75 and the same fertilizer placement under Deep band, Side band, and Broadcast (Figure 7B–D) at 115 DAS, except a significantly greater seed N concentration in N187 than in N112 for the same P45 under Deep band (Figure 7B). Seed N concentration was similar among different P fertilization rates of P45, P60, and P75 for the same N fertilization rate either at N112, N150, or N187 and the same fertilizer placement under Deep band, Side band, and Broadcast at 115 DAS (Figure 7B–D), except a significantly greater seed N concentration in P75 than in P45 for N112 under Deep band (Figure 7B,D).

#### 2.4.2. Seed N Accumulation

Seed N accumulation (mg plant^−1^) in the no-fertilization control was similar among different fertilizer placements under the Deep band, Side band, and Broadcast (Figure 7E), regardless of NP fertilization rate. 

No significant difference was observed in seed N accumulation among different fertilizer placements (Deep band, Side band, and Broadcast) under the same N and P fertilization rates (N112P45, N150P45, N187P45, N112P60, N150P60, N187P60, and N112P75; Figure 7F–H), except a significantly greater seed N accumulation in Deep band than in Side band and Broadcast under N150P75 (Figure 7H), as well as in Deep band than in Broadcast under N187P75 (Figure 7H). Seed N accumulation was similar among different N fertilization rates of N112, N150, and N187 for the same P fertilization rates either at P45, P60, or P75 and the same fertilizer placement under Deep band, Side band, and Broadcast (Figure 7F–H), except a significantly greater seed N accumulation in N187 than in N112 for P75 under Deep band (Figure 7H).

Seed N accumulation was similar among different P fertilization rates of P45, P60, and P75 with the same N fertilization rates at N112, N150, and N187 and the same fertilizer placement under Deep band, Side band, and Broadcast (Figure 7F–H), except a significantly great seed N accumulation in P75 than in P45 for N150 under Deep band (Figure 7F,H).

### 2.5. Total Plant N Accumulation

Total plant (leaf + stem + seed + root) N accumulation (mg plant^−1^) in the control no-fertilization was similar between three different fertilizer placements of Deep band, Side band, and Broadcast at 115 DAS (Figure 7I), no matter whether of NP fertilization rate. 

Among different fertilizer placements for the same N and P fertilization rate, total plant N accumulation was significantly greater in the Deep band than in the Side band and Broadcast under N150P60, N187P60, N112P75, and N150P75, (Figure 7K,L). Likewise, significantly greater total plant N accumulation was observed in the Deep band than in Broadcast under N112P45, N150P45, N112P60, and N187P75 (Figure 7J,K). However, total plant N accumulation was not significantly different among the Deep band, Side band, and Broadcast under N187P45 (Figure 7J). Total plant N accumulation among different N fertilization rates of N112, N150, and N187 for the same P45 fertilization rate was significantly greater in N187 than in N112 under the Deep band and Side band. However, no significant difference was observed among different N fertilization rates for P45 under Broadcast (Figure 7J). Meanwhile, among different N fertilization rates for the same P60 fertilization rate, total plant N accumulation was significantly greater in N187 than in N112 under the Broadcast and Side band, as in N187 and N150 than in N112 under the Deep band (Figure 7K). In addition, total plant N accumulation among different N fertilization rates for the same P75 fertilization rate was significantly greater in N187 and N150 than in N112 under the Deep band, as in N187 than in N112 under the Side band, and in N187 ≈ N150 ≈ N112 under Broadcast (Figure 7L).

Among different P fertilization rates for the same N fertilization rate and same fertilizer placement, total plant N accumulation was significantly greater in P75 and P60 than in P45 for N112 under Broadcast, and for N150 and N187 under Deep band (Figure 7J–L), as well as in P75 than in P45 for N150 under Broadcast (Figure 7J,L). However, no significant difference in total plant N accumulation was observed among P75, P60, and P45 for N187 under Broadcast, for N112, N150, and N187 under Side band, and N112 under Deep band (Figure 7J–L).

### 2.6. Total Soil N Concentration

Total soil N concentration (g kg^−1^) in the no-fertilization control decreased with plant aging, but there was no significant difference between 60 DAS and 115 DAS (Figure 8A,E,I).

Total soil N concentrations were similar between 60 DAS and 115 DAS for all the same N and P fertilization rates (N112P45, N112P60, N112P75, N150P45, N150P60, N150P75, N187P45, N187P60, or N187P75) under both Broadcast and Side band fertilizer placement (Figure 8B–D,F–H,J–L). In contrast, total soil N concentration was significantly greater in 60 DAS than in 115 DAS for all the same N and P fertilization rates under Deep band. 

Significantly greater total soil N concentration among different fertilizer placements with the same N and P fertilization rate was observed in the Deep band than in the Side band and Broadcast under N112P45, N150P45, N187P45, N112P60, N150P60, N187P60, N112P75, N150P75, and N187P75 fertilization rates at 60 DAS, while total soil N concentration was not significantly different between Deep band, Side band, and Broadcast under the same N and P fertilization rates at 115 DAS (Figure 8D,F–H,J–L). 

Total soil N concentrations were similar among different N fertilization rates of N112, N150, and N187 with the same P45 fertilization rate under Deep band, Side band, and Broadcast at 60 DAS and 115 DAS, except for a significantly greater total soil N concentration in N187 than in N112 under Broadcast at 60 DAS (Figure 8B–D). Moreover, total soil N concentrations were similar among different N fertilization rates of N112, N150, and N187 with the same P60 fertilization rate under the Deep band, Side band, and Broadcast at 60 DAS and 115 DAS (Figure 8F–H). In addition, total soil N concentrations were also similar among different N fertilization rates of N112, N150, and N187 with the same P75 fertilization rate under Broadcast at 60 DAS and under Deep band, Side band, and Broadcast at 115 DAS, while significantly greater total soil N concentration was observed in N187 than N112 under the Deep band and Side band at 60 DAS (Figure 8J–L).

Significantly greater total soil N concentrations among different P fertilization rates for the same N fertilization rates and same fertilizer placements were observed in P75 and P60 than in P45 for N112 under Broadcast at 60 DAS (Figure 8B,F,J), in P75 than in P60 and P45 for N187 under Deep band and Side band at 60 DAS (Figure 8D,H,L) and for N150 under the Side band and Broadcast at 60 DAS (Figure 8C,G,K), and in P75 than in P45 for N187 under Broadcast at 60 DAS (Figure 8D,H,L). In addition, total soil N concentrations were similar among different P fertilization rates of P45, P60, and P75 for the same N112 under Broadcast at 115 DAS and under the Side band and Deep band at 60 DAS and 115 DAS (Figure 8B,F,J), for N150 under Broadcast and Side band at 115 DAS and under the Deep band at 60 DAS and 115 DAS (Figure 8C,G,K), as well as for N187 under Broadcast, Side band, and Deep band at 115 DAS (Figure 8D,H,L).

### 2.7. Nitrogen Use Efficiency

Deep fertilizer placement significantly enhanced greater nitrogen agronomy efficiency (NAE), and partial factor productivity for the applied N (PFP_N_) than Broadcast and Side band under the same N150P75 and N187P75, as well as nitrogen use efficiency (NUE) under the same N150P60, N187P60, N150P75, N187P75 (Table 1). Moreover, significantly greater NUE was enhanced in N112 than in N150 and N187 for the same P60 or P75 under Broadcast; as well as PFP_N_ for the same P45 or P75 under Broadcast and Deep band, and also for the same P60 or P75 under Side band (Table 1). In addition, P75 and P60 as compared to P45 significantly increased NAE, NUE, and PFP_N_ than P45 for the same N150 and N187 under Deep band (Table 1).

### 2.8. Relationships between Concentrations of Tissue N or Soil Total N and Plant N Accumulations

Plant tissue N accumulations positively correlated with tissue N concentrations in leaf, stem, and root (*r*^2^ = 0.81–0.84, *p* = 0.01–0.001) at the VT (tasseling, at 60 DAS) growth stage (Figure 9A–C), Similarly, leaf, stem, and root N concentrations or accumulations positively correlated with total soil N concentrations (*r*^2^ = 0.57–0.69, *p* = 0.01–0.001) at 60 DAS (Figure 9D–I). 

Leaf, stem, root, seed, and total plant (leaf + stem + seed + root) N accumulations positively correlated with their tissue N concentrations (*r*^2^ = 0.52–0.88, *p* = 0.01–0.001) at the R6 (physiological maturity or harvest, at 115 DAS) growth stage (Figure 10A–E). Likewise, leaf, stem, seed, root, and total plant (leaf + stem + seed + root) N concentrations or accumulations positively correlated with total soil N concentrations (*r*^2^ = 0.23–0.66, *p* = 0.01–0.001) at 115 DAS (Figure 10F–O).

## 3. Discussion

Numerous studies have shown that either Side band or Deep band fertilizer placement can maintain a better N and P availability in the root zone for a long-term crop growth requirement, thereby improving nutrient uptake, NUE, and seed yield [23,24,25]. In a 2-year (2018 and 2019) greenhouse pot experiment using nine different combinations of N and P fertilization rates (N112P45, N112P60, N112P75, N150P45, N150P60, N150P75, N187P45, N187P60, and N187P75 kg ha^−1^), this present study verified such growth advantages or differences between three contrasting of Broadcast, Side band, and Deep band fertilizer placements over their whole growth stage of two maize hybrid verities (Xida-789 and Xida-211, averaged as no differences in variety). 

### 3.1. Greater N Concentration under Deep Band 

Among fertilizer placements for the same N and P fertilization rate, leaf N concentrations at 60 DAS were considerably higher under Deep band than under Side band and Broadcast by 7.19% and 10.21% under N150P60, 5.76% and 8.87% under N18760, 6.69% and 10.40% under N15075, and 7.26% and 8.68% under N187P75, respectively (Figure 2G,H,K,L). Compared to Side band and Broadcast, stem N concentrations in Deep band at 45 or 60 DAS were also increased up to 7.55–10.47 or 10.85–11.86% under N150P75, and to 8.55–10.89% or 10.13–11.38% under N187P75 (Figure 4K,L). These results indicated that the deep band placement substantially increased plant N uptake than the side band and broadcast did. It should be known that broadcast does not often ensure proper plant nutrition since the placement of fertilizer by broadcast is beyond the reach of root systems [20]. In contrast, deep placement provides more nutrients in the root zone and rhizosphere [26], hence facilitating nutrients to be better distributed in the area of root systems and thereby effectively taken up by roots [27]. Furthermore, at the early growth stage of maize, deep placement provides immediate nutrient access to the newly emerged roots and increases the concentration of immobile nutrients (P and K) near the maize rooting zone [24]. Moreover, deep placement maintains high levels of available nutrients closer to the root, which stimulates root growth in the localized nutrient-rich pitches and in the immediate vicinity of localized nutrient application [28]. In addition, it has also been reported that root growth was better in soil with high NO_3_^−^-N than those with low NO_3_^−^-N under deeper N application [29]. 

It has been shown that root zone fertilization considerably increases N concentration and available N for a longer time, ensuring the continuous supply of high N to maize demands over a period of 90 days [27]. Crops can obtain ≤50% of their N from deeper soil layers when N availability is limited in topsoil [30], as a result of improved N uptake by placing N fertilizer in a deeper and wetter soil layer [20]. Our results are in accordance with previous studies. For example, as an optimal fertilizer placement, an N100 + P30.8 kg ha^−1^ rate at 5 and 10 cm soil depth resulted in significant increases in both N uptake and utilization in maize [20]. Nitrogen concentrations in maize grains or N derived from fertilizer (Ndff) were significantly increased from 12.8–13.1 g kg^−1^ or 17.5% under broadcast to 13.4–13.9 g kg^−1^ or 22.9% under 12 cm deep placement at N135 and N180 kg N ha^−1^ rate [26]. Furthermore, a meta-analysis with 474 observations in the USA showed an average of a 5.2% increase in maize yield by N and P placement at a 5 cm depth close to a 5 cm seeding zone at sowing [24]. By contrast, neither N nor P alone, only the NP combination at 10 cm placement increased the grain yield of maize [31]. 

An optimal rate of N and P placement at 15–30 cm depth improved root growth, N and P accumulations, and nutrient use efficiency of *Rosa multiflora* [32]. Compared to three surface doses for a total of 225 kg N kg ha^−1^ (urea, 40%, 30%, and 30% at planting, tillering, and jointing stage), one-time 5 cm or 10 cm depth N placement from a 5 cm rice root zone accumulated greater N in leaf and grain of rice [33]. Studies also showed that a combined N + P fertilization produced a positive response to nutrient uptake. For example, the deep placement of (NH_4_)_2_SO_4_ and P, i.e., 46.2 of N + 74.5 of P kg ha^−1^ at sowing and the top dressing of N118.8 + P40.5 kg ha^−1^ through Broadcast at the jointing stage considerably improved N and P uptake by maize [34]. In addition, fertilizer application at 10 cm depth substantially reduced N and P losses while promoting the utilization efficiency of N and P fertilizer, such improvements could provide a theoretical base for decreasing future N and P losses and applying mechanized fertilization [35].

### 3.2. Greater Soil Total N Concentration under Deep Band 

In the present study, among different fertilizer placements for the same N and P fertilization rate, total soil N concentration was significantly greater in the Deep band than in the Side band and Broadcast by 11.44% and 17.10% under N112P45, 12.52% and 15.30% under N150P45, 12.07% and 13.37% under N187P45, 9.59% and 10.77% under N112P60, 10.73% and 11.86% under N150P60, 11.11% and 11.41% under N187P60, 8.68% and 9.15% under N112P75, and 12.24% and 15.93% under N187P75 at 60 DAS, respectively (Figure 9B–D,F–H,J–L). Similar to our findings, Wang et al. [36] have reported that N placement at 10 cm depth under N125 and N195 kg ha^−1^ fertilization rate enhanced higher soil N concentration at the mid-tillering stage than at the heading stage of rice. They further indicated that the soil total N concentration at 0–20 cm soil depth under N125 and N195 was 2.17 and 2.28 mg g^−1^ in the mid-tillering stage, and 2.04 and 2.15 mg g^−1^ in the heading stage, respectively. Soil NO_3_^−^–N was effectively promoted in the soil profile under deep fertilizer placement [37]. In addition, soil NO_3_^−^–N under deep fertilizer placement was lower at 25 and 35 cm depth than that at 5 and 15 cm [38]. Moreover, deep fertilizer placement reduces N losses, i.e., N_2_O emission, NH_3_ volatilization, and NO_3_ leaching [39,40], and prolongs the availability of N concentration in the soil for root uptake during the crop growth cycle, and thus promotes plant growth, increase grain yield and NUE [15,30,41]. According to Jiang et al. [21], compared with the split surface broadcast, the N recovery under 12 cm depth of fertilizer placement was considerably higher by 30.0% under N135 kg ha^−1^, 21.9% under N180 kg ha^−1^ and 23.6% under N225 kg ha^−1^, while N losses were reduced by 24.2% under N135, 20.8% under N180, and 11.2% under N225, respectively. Wu et al. [39] also showed that 25 cm deep fertilizer placement could reduce gaseous N loss by decreasing NH_4_^+^-N concentration in the soil surface thus ultimately increasing N uptake, NUE, and maize yield.

### 3.3. Greater N Accumulation and Yield Production under Deep Band 

In the present study, among different fertilizer placements for the same N and P fertilization rate, the Deep band significantly increased leaf, stem, and root N accumulations than the Side band and Broadcast under N150P60, N187P60, N150P75, and N187P75 at 45 DAS, 60 DAS or 115 DAS (Figure 3, Figure 5 and Figure 8G,H,K,L); seed N accumulation under N150P75 at 115 DAS (Figure 6G); and total plant (leaf + stem + seed + root) N accumulation under N150P60, N187P60, N112P75, N150P75, and N187P45 at 115 DAS (Figure 6I–L). The possible explanation for higher N accumulation in different plant organs with Deep band could be that (i) deep fertilizer placement improves root growth and distribution and hence enlarges the size of the root system, which thereby increases nutrient uptake accumulation and thus biomass production [27,28,33], (ii) as often the major inorganic N form, NO_3_^−^ is mobile and its uptake by plant root depends on soil moisture. Broadcasting N fertilizer in a drier surface soil therefore limits N availability for plant uptake, while the N fertilizer under deep placement is applied in a wetter deep soil where the mobility of NO_3_^−^ is increased and so does the N uptake by plant roots [42], (iii) deep fertilizer application increases the nutrient concentration in the root zone, preserves nutrient for a longer period and reduces microbial competition with plants and thereby minimizes N losses through NO_3_ leaching, NH_3_ volatilization, and N_2_O emission. All of these three factors could increase plant NUE, nutrient uptake, and seed yield [26,29,43]. Moreover, an optimum N and P fertilization rate coupled with deep fertilizer placement, especially at the initial growth stage, could promote crop growth and nutrient accumulation while enhancing higher nutrient use efficiency, productivity, and profitability [15,20,27,42]. With 602 datasets from 33 studies between 1982 and 2015, a meta-analysis compared nutrient utilization between the fertilizer placement methodologies including banding, split, subsurface, or broadcast fertilizer placements [15]. These results also showed lower concentrations and accumulations of plant N, P, and grain protein under broadcast. Under deep band placement, both NH_3_ volatilization and N_2_O emission decreased with increasing soil depth because of NH_4_^+^-N fixation by the negative charge of soil colloids and restriction of NH_4_^+^ movement by relatively higher clay in the upper soil [39]. With a decreased N_2_O emission, deep fertilizer placement at 20 cm soil depth increased grain N by 3.6% and 2.5% than shallow placement at 7 cm and mixed placement (half fertilization rate at 7 cm and 20 cm), respectively [39]. In addition, deep placement also showed economic and environmental advantages. For example, compared with root zone fertilization (i.e., side band placement), N split broadcast required additional labor, caused N to leach into the environment, and decreased maize yield [27]. Moreover, there was a tradeoff between NH_3_ emissions and economic indicators under deep fertilizer placement in a rice field crop in China. For instance, deep fertilizer placement considerably reduced NH_3_ emission by 78.1% and enhanced total output and net economic return by 13.6% and 11.5%, respectively [6]. Likewise, both N135 and N180 kg ha^−1^ rates at 12 cm soil depth considerably increased N accumulation in different maize tissues than the same rates under split surface broadcast [26], N fertilizer placement at the depths of 5, 10, and 15 cm considerably increased N uptake and seed yield [32], and root zone fertilization of urea at either 10 cm or 5 cm depth enhanced greater accumulations of N, P and potassium in leaf, stem, and seed of rice in sandy and loam soils [33]. Hence, an adequate fertilization rate and fertilizer placement methodology are therefore timely needed to meet plant N demands, minimize nutrient losses, and improve crop production [44]. 

The N and P interaction is one of the most important interactions in enhancing crop production [22], which promotes root mass, root length, and the number of root tips, resulting in the capturing and acquisition of water and nutrients including higher N uptake and thus seed yield [32,45,46]. In a situation where P is limiting, the sole N application greatly reduced grain yield, whereas the combined N and P supply considerably increased soil NO_3_-N utilization and grain yield [47]. In another long-term experiment from 1992 to 2010, Schlegel et al. [31] showed that the fertilization of N or P alone increased maize grain yield by 103% and 20%, respectively, over control, while the combined N and P application increased grain yield up to 225% over the control. Therefore, a combined N and P fertilization is a major strategy to improve crop productivity [22].

The current study showed that leaf and stem N accumulations at 45 DAS, 60 DAS or 115 DAS, root N accumulation at 60 DAS and 115 DAS, and total plant N accumulations at 115 DAS were considerably greater in P75 and P60 than in P45 for the same N150 and N187 under Deep band, respectively (Figure 3C,G,K, Figure 5C,D,G,H,K,L, Figure 6J–L and Figure 8C,D,G,H,K,L). Indeed, phosphorus plays a significant role in nutrient management and enhances higher crop yield as it is a modifying enzyme in phosphorylation that regulates metabolic processes and is necessary for cell signaling and division [48]. According to Gu et al. [49], 90 kg N ha^−1^ plus120 or 135 kg P ha^−1^ increases the activities of nitrate reductase, glutamine synthase, and glutamate synthase and thereby improves the assimilation of NO_3_-N and NH_4_^+^-N. They also reported that the same N and P fertilization rates improve the activity of glutamic-oxalacetic transaminase and glutamic-pyruvate transaminase which in turn increased the content of amino acids. In contrast, low P supply decreases leaf energy availability, nitrate reductase activity, and glutamine synthesis, and thus amino acid metabolism and N-assimilation [50,51].

### 3.4. Greater N Use Efficiency under Deep Band

In modern agriculture, the most important priorities are given to improve the utilization efficiency of nitrogen from mineral fertilizers [52]. The NAE (N agronomy efficiency), NUE (N use efficiency), and PFP_N_ (partial factor productivity for the applied N) are commonly characterized for fertilizer use efficiency [15,27,53]. Fertilizer management strategies by optimizing the right rate, right source, right time, and right application method have obtained higher yield, N efficiency, and environmental quality [54]. In the current study, Deep band significantly increased NAE, NUE, and PFP_N_ more than Broadcast and Side band (Table 1). It could be attributed that deep fertilizer placement provided easy N and P access to the maize roots which increased nutrient use efficiencies while reducing nutrient losses [34]. Similar to our findings, Sandhu et al. [55] showed that application of N90 and N120 at 12–15 cm soil depth significantly increased NAE and NUE in maize as compared to broadcast. A one-time root zone fertilization at 6, 9, 12, or 15 cm particularly at 12 cm soil depth improved NAE and NUE than a split surface broadcast [27]. 

Likewise, N fertilizer placement at the depth of 5 cm, 10 cm, and 15 cm utilized applied N fertilizer more efficiently, and improved NAE than N split application to the soil surface [32]. Moreover, NAE in maize was increased from 3.5 kg grain kg^−1^ N under the sole 187.5 kg N ha^−1^ to 16.3 kg grain kg^−1^ N under the same N rate plus 41 kg P ha^−1^ in China [56]. Compared with a sole N fertilization at four maize plantation sites across China, NUE was increased from 20% to 45% under N plus P fertilization [57]. In comparison to the farmer’s surface fertilization in sandy and loam soils, root zone urea fertilization at 10 cm or 5 cm depth enhanced greater N apparent recovery efficiency (53.1% vs. 27.5%) [33]. In addition, compared to broadcast, 12 cm root zone fertilization considerably increased ^15^N recovery and N apparent recovery in maize by 21.9–30.0% and 14.3–37.8%, respectively, while decreased N losses by 11.2–24.2%, respectively [21].

Use efficiencies of N and P generally have positive relationships with grain yields under appropriate fertilizer management [15,16,53,58,59]. However, excess fertilizer use coupled with inappropriate fertilizer placement methods could increase nutrient losses. In the current study, a lower N112 fertilization rate considerably increased NUE and PFP_N_ than both higher N150 and N187 fertilization rates (Table 1). Studies also showed that the NUE and PFP_N_ were generally higher under a lower N fertilization rate and decreased with increasing N fertilization rate [55,60]. A higher NUE and PFP_N_ under a lower N fertilization rate could be the result of better utilization of N uptake in maize grains and lower N loss to the environment, while A lower NUE and PFP_N_ under a higher N fertilization rate could be the result of a higher N loss from NH_3_ volatilization, leaching, and denitrification [15,16,60].

### 3.5. Variations in Relationships between Concentrations of Tissue N, Soil N, and Plant N Accumulations

In the present study, plant tissue N accumulation positively correlated with tissue N concentration (Figure 9A–C and Figure 10A–D), while tissue N concentration and accumulation in leaf, stem, seed, root, and total plant positively correlated with total soil N concentration (Figure 9D–I and Figure 10E–N). There is often a positive relationship between plant N uptake and tissue N concentration and they are linearly increased with increasing soil N concentration [58,61]. Generally, an increase of plant tissue N with soil N will ultimately increase plant N accumulation and grain yield [58]. Deep N and P fertilizer placement increases nutrient availability in the maize root zone for a longer period of growth requirement [62], which can promote root length, proliferation, and penetration capacity, as well as water and N uptake, and thereby plant tissue N accumulation and biomass production [62,63]. Our results are in accordance with these previous studies, e.g., Fageria et al. [59] reported positive relationships between soil NO_3_^−^ and plant tissue N. Jiang et al. [21] showed that maize tissue N accumulation increased with the increasing N fertilization rates from N135, N150 to N180 kg ha^−1^. Asibi et al. [64] reviewed that N accumulation in the aboveground biomass increased with increasing soil N concentration. Singh et al. [65] indicated that leaf N concentration of common trees (*Quercus leucotrichophora*, *Pinus roxburghii*, *Cupressus torulosa*, *Alnus nepalensis*, and *Populus ciliata*) and shrub species (*Desmodium elegans* and *Crataegus crenulata*) were positively correlated with soil N concentration. Moreover, a positive relationship was also observed between plant tissue N accumulation and seed yield or biomass production. For example, a significant linear correlation was observed between plant N accumulation and biomass production [66], N accumulation and total aboveground biomass and seed yield [67,68], and maize, rice, or wheat tissue N accumulation and their grain yield [69,70]. 

## 4. Materials and Methods

### 4.1. Experimental Site 

A two-year pot experiment was carried out with nearly natural light and temperature under a rain-proof glass greenhouse (29 May to 15 September 2018, and 15 May to 28 August 2019) at the National Monitoring Base for Purple Soil Fertility and Fertilizer Efficiency (29°48′ N, 106°24′ E, 266.3 m above sea level) on the campus of Southwest University Chongqing, China. The region has a humid subtropical climate with a mean annual temperature of 9.1 °C and 10.8 °C in winter and 26.0 °C and 25.0 °C in summer of 2018 and 2019, and a mean precipitation of 143.2 mm and 157.1 mm during May to September of the maize growing season in 2018 and 2019, respectively (Table 2). The typical soil in this area has been developed from gray-brown-purple sand shale parent materials in the Mesozoic Jurassic Shaxi Temple Group and is categorized as Eutric Regosol (FAO Soil Classification System) and belongs to the Cabhaplic Stagnic Anthrosols. 

### 4.2. Experimental Design and Treatments

The experiments were arranged in a random split-plot design with three fertilizer placement methodologies (Broadcast, Side band, and Deep band) as the main factor and a combination of four N and P fertilization rates (No-NP, 112, 150, 187 kg N ha^−1^, and 45, 60, 75 kg P ha^−1^) as the sub-factor. As a result, a total of 11 NP combined fertilization treatments were formed as No-NP, N112P45, N112P60, N112P75, N150P45, N150P60, N150P75, N187P45, N187P60, and N187P75. Studies have shown that N or P in the form of NH_4_^+^ or PO_4_^3−^ is suitable for fertilizer placement experimentation [15]. As a result, commercial urea (46% N) and calcium superphosphate (P_2_O_5_ ≥ 12%) with potassium chloride (40 kg ha^−1^, 52% K) were once applied as basal fertilizers for one-time fertilization prior to seeding. 

Each fertilization treatment had three replicates for a total of 33 replicates or pots to each fertilization replacement methodology (see Figure 11 for details of the experimental set-up). An experimental pot (length × width × height = 57 × 23 × 27 cm) was filled with 26 kg purple soil. Prior to the experiment, this soil (collected from 0–20 cm depth within the above-mentioned National Monitoring Station Base for Purple Soil Fertility and Fertilizer Efficiency) had 14% sand, 48% silt, and 38% with a bulk density of 1.37/cm^3^, pH (1:2.5; H_2_O) of 6.9, 7.40 g soil organic carbon kg^−1^, 0.70 g total N kg^−1^, 0.42 g total P kg^−1^, 10.16 g total potassium kg^−1^, 81.90 mg available N kg^−1^, 15.70 mg available P kg^−1^, and 176 mg available K kg^−1^, respectively. 

### 4.3. Crop Management

The maize hybrid varieties of Xida-789 and Xida-211 were grown in 2018 and 2019, respectively. Seeds of maize were surface sterilized with 10% H_2_O_2_ for 20 min, thoroughly rinsed with sterile water, and then pre-germinated on sterilized moist filter paper at 25/20 °C (day/night) for 36 h. Five seeds were sown and two healthy seedlings were kept per pot after ten days of germination. During the growing season, experimental plants were monitored on a daily basis for any possible biotic and abiotic risks. Soil moisture during the whole growth period was maintained at 70% field water-holding capacity. Weeding was carried out at 15 DAS and 45 DAS, respectively. During the tasseling stage, the Philippine downy mildew (pale yellow to whitish discolorations) was controlled with 1/1000 diluted 25% Metalaxyl solution. 

### 4.4. Plant and Soil Sampling and Chemical Assay

Plant aboveground biomass (leaf and stem) were harvested at the V8 (eight-leaf, 45 DAS) stage, VT (tasseling, 60 DAS) stage, and R6 (physiological maturity or harvest, 115 DAS) stage while root biomass was harvested at 60 DAS and 115 DAS. The harvested plants were partitioned into leaves, stems, seeds, and roots (carefully washed with tap water), and then oven dried at 60 °C until a constant weight was reached. After drying, leaf, stem, seed, and root dry weights were recorded. The oven-dried plant tissues were grounded and passed through a 1 mm screen and then digested with 98% sulfuric acid and 30% hydrogen peroxide for N analysis according to the Micro-Kjeldahl method [71]. Soil samples were air dried until a constant weight, grounded and passed through a 0.25 mm screen, and then digested with 98% sulfuric acid and a catalyst mixture of potassium sulfate (K_2_SO_4_) and cupric sulfate (CuSO_4_)_._ Total soil N concentration was determined according to the Micro-Kjeldahl method [71]. Plant tissue N accumulation was calculated by multiplying leaf, stem, seed, and root N concentration with leaf, stem, seed, and root biomass, respectively, whereas total plant N accumulation was calculated as the sum of leaf + stem + seed + root N accumulation.

### 4.5. Calculations of Agronomy Nitrogen Use Efficiency, Nitrogen Use Efficiency, and Partial Factor Productivity of N Fertilizer

Nitrogen agronomy efficiency (NAE) is calculated by the following equation:

NAE (kg grain kg^−1^ N) = (YN−Y0)/AN × 100%
where YN or Y0 is the grain yield under a specific N or no-N fertilization (control), and AN is the total applied N from the N fertilization.

2.Nitrogen use efficiency (NUE) is calculated by the following equation:

NUE (%) = (UN−U0)/AN × 100
where UN or U0 is total N accumulation (leaf + stem + seed + root) under a specific N or no-N fertilization (control), and AN is the total applied N from the N fertilization.

3.The partial factor productivity of nitrogen fertilizer (PFP_N_) is calculated by the following equation:

PFP_N_ (kg grain kg^−1^ N) = YN/AN
where YN is the grain yield and AN is the total N accumulation under a specific N fertilization.

### 4.6. Statistical Analyses 

Statistical analyses were performed using SPSS 24.0 software (SPSS Inc., Chicago, IL, USA). Data were expressed as means ± standard error (*n* = 6). One-way ANOVA was performed to test the treatment (fertilizer placement method, N, P fertilization levels, and growth stage) difference, and significant differences among treatments were compared by the Duncan’s multiple range test at *p* < 0.05. An OriginPro2023b (version-10.05) software (Origin Lab Corp., Northampton, MA, USA) was used for graphs, correlation, and analysis. 

## 5. Conclusions

Synchronized application of N150P60, N187P60, N150P75, and N187P75 with deep fertilizer placement increased N availability in the root zone and promoted root growth and nutrient uptake, thereby increasing N concentration and N accumulation in maize’s aboveground organs. Hence, Deep band in this study emerged as the best placement for N and P fertilizers than Side band and Broadcast. Among different N rates, N187 and N150, rather than N112, with the same P60 and P75 rates under Deep band significantly increased leaf, stem, root, and total plant N accumulation. In contrast, NUE and PFP_N_ were significantly greater in N112 than in N150 and N187 for the same P60 or P75, irrespective of fertilizer placement. Among different P rates, P75 and P60, rather than P45, with the same N187 and N150 under Deep band considerably increased N concentration and accumulation in different maize tissues, and NAE, NUE, and PFP_N_. Although results from this pot experiment need to be further studied in the field, a combined N P application under Deep band placement presents an important management practice to improve soil N availability, tissue N accumulation, N use efficiency, and thereby yield production in maize plantation, particularly for small-holder farmers around the world. Nevertheless, since soil and plant growth conditions in greenhouses are largely different from those in the field, useful information based on pot experiments, must be cautiously analogized or referred to field practice, and future field experiments are therefore timely required to further evaluate the effect of N and P fertilization under placement methodology on promoting crop production.

## Figures and Tables

**Figure 1 plants-12-03870-f001:**
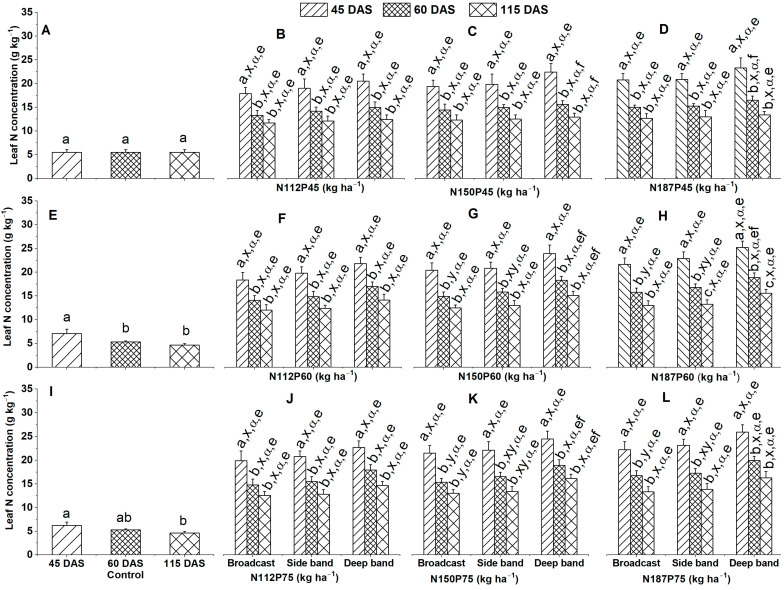
Effects of nitrogen fertilization rates and fertilizer placements on leaf N concentrations of maize in the V8 (eight-leaf) growth stage at 45 days after sowing, VT (tasseling) stage at 60 days after sowing, and the R6 (physiological maturity or harvest) stage at 115 days after sowing. Figures (**A**,**E**,**I**) represent no-fertilization control treatment; (**B**–**D**) showed difference between different N fertilization rates and fertilizer placements for the same P45 fertilization rate; (**F**–**H**) between different N fertilization rates and fertilizer placements for the same P60 fertilization rate; (**J**–**L**) between different N fertilization rates and fertilizer placements for the same P75 fertilization. Since there were no significant differences between varieties (Xida-789 and Xida-211) and years (2018 and 2019), data are combined under three fertilizer placements for the control treatments and under the same N and P fertilizer for the same fertilizer placement. Data (means ± SE, *n* = 6) followed by different letters indicate significant differences between different growth days for the same N and P fertilization rate and same fertilizer placement or between 45 DAS, 60 DAS, and 115 DAS under the control (a, b, c), between different fertilizer placements for the same growth day and same N and P fertilization rate (x, y), between different N fertilization rates with the same P fertilization rate for the same fertilizer placement and plant growth day (α), and between different P fertilization rates with the same P fertilization rate for the same fertilizer placement and plant growth day (e, f) at *p* < 0.05. Abbreviations: DAS, days after sowing; N, nitrogen; P, phosphorus.

**Figure 2 plants-12-03870-f002:**
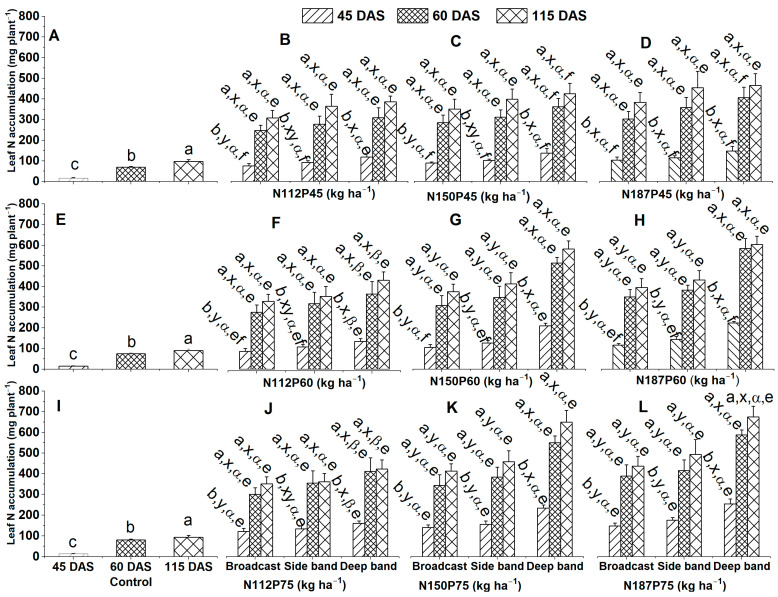
Effects of nitrogen fertilization rates and fertilizer placements on leaf N accumulations of maize in the V8 (eight-leaf) growth stage at 45 days after sowing, VT (tasseling) stage at 60 days after sowing, and the R6 (physiological maturity or harvest) stage at 115 days after sowing. Figures (**A**,**E**,**I**) represent no-fertilization control treatment; (**B**–**D**) showed difference between different N fertilization rates and fertilizer placements for the same P45 fertilization rate; (**F**–**H**) between different N fertilization rates and fertilizer placements for the same P60 fertilization rate; (**J**–**L**) between different N fertilization rates and fertilizer placements for the same P75 fertilization. Since there were no significant differences between varieties (Xida-789 and Xida-211) and years (2018 and 2019), data are combined under three fertilizer placements for the control treatments and under the same N and P fertilizer for the same fertilizer placement. Data (means ± SE, *n* = 6) followed by different letters indicate significant differences between different growth days for the same N and P fertilization rate and same fertilizer placement or between 45 DAS, 60 DAS, and 115 DAS under the control (a, b), between different fertilizer placements for the same growth day and same N and P fertilization rate (x, y), between different N fertilization rates with the same P fertilization rate for the same fertilizer placement and plant growth day (α, β), and between different P fertilization rates with the same N fertilization rate for the same fertilizer placement and plant growth day (e, f) at *p* < 0.05. Abbreviations: DAS, days after sowing; N, nitrogen; P, phosphorus.

**Figure 3 plants-12-03870-f003:**
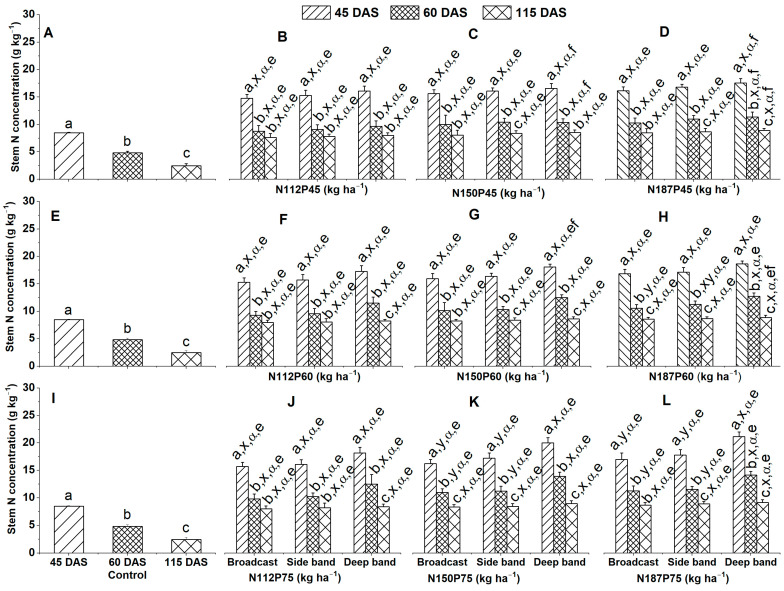
Effects of nitrogen fertilization rates and fertilizer placements on stem N concentrations of maize in the V8 (eight-leaf) growth stage at 45 days after sowing, VT (tasseling) stage at 60 days after sowing, and the R6 (physiological maturity or harvest) stage at 115 days after sowing. Figures (**A**,**E**,**I**) represent no-fertilization control treatment; (**B**–**D**) showed difference between different N fertilization rates and fertilizer placements for the same P45 fertilization rate; (**F**–**H**) between different N fertilization rates and fertilizer placements for the same P60 fertilization rate; (**J**–**L**) between different N fertilization rates and fertilizer placements for the same P75 fertilization. Since there were no significant differences between varieties (Xida-789 and Xida-211) and years (2018 and 2019), data are combined under three fertilizer placements for the control treatments and under the same N and P fertilizer for the same fertilizer placement. Data (means ± SE, *n* = 6) followed by different letters indicate significant differences between different growth days for the same N and P fertilization rate and same fertilizer placement, or between 45 DAS, 60 DAS, and 115 DAS under the control (a, b, c), between different fertilizer placements for the same growth day and same N and P fertilization rate (x, y), between different N fertilization rates with the same P fertilization rate for the same fertilizer placement and plant growth day (α), and between different P fertilization rates with the same N fertilization rate for the same fertilizer placement and plant growth day (e, f) at *p* < 0.05. Abbreviations: DAS, days after sowing; N, nitrogen; P, phosphorus.

**Figure 4 plants-12-03870-f004:**
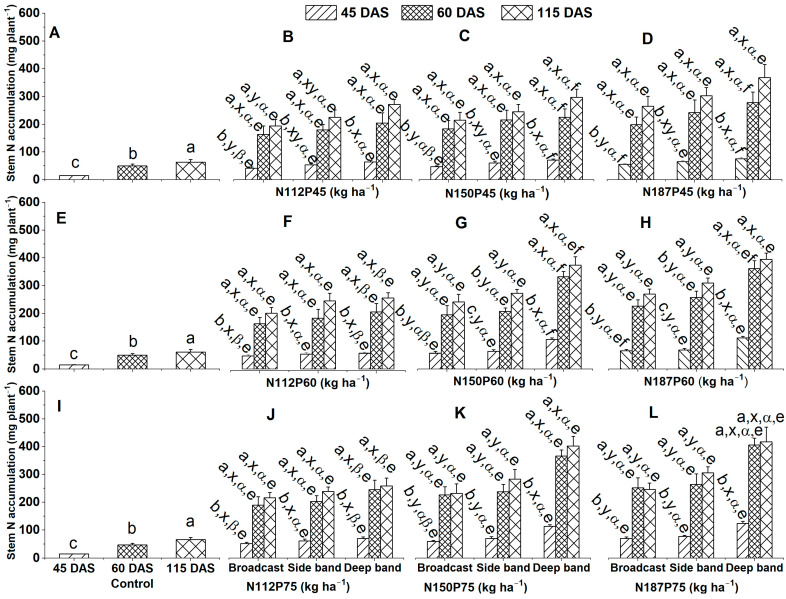
Effects of nitrogen fertilization rates and fertilizer placements on stem N accumulations of maize in the V8 (eight-leaf) growth stage at 45 days after sowing, VT (tasseling) stage at 60 days after sowing, and the R6 (physiological maturity or harvest) stage at 115 days after sowing. Figures (**A**,**E**,**I**) represent no-fertilization control treatment; (**B**–**D**) showed difference between different N fertilization rates and fertilizer placements for the same P45 fertilization rate; (**F**–**H**) between different N fertilization rates and fertilizer placements for the same P60 fertilization rate; (**J**–**L**) between different N fertilization rates and fertilizer placements for the same P75 fertilization. Since there were no significant differences between varieties (Xida-789 and Xida-211) and years (2018 and 2019), data are combined under three fertilizer placements for the control treatments and under the same N and P fertilizer for the same fertilizer placement. Data (means ± SE, *n* = 6) followed by different letters indicate significant differences between different growth days for the same N and P fertilization rate and same fertilizer placement, or 45 DAS, 60 DAS, and 115 DAS under the control (a, b, c), between different fertilizer placements for the same growth day and same N and P fertilization rate (x, y), between different N fertilization rates with the same P fertilization rate for the same fertilizer placement and plant growth day (α, β), and between different P fertilization rates with the same N fertilization rate for the same fertilizer placement and plant growth day (e, f) at *p* < 0.05. Abbreviations: DAS, days after sowing; N, nitrogen; P, phosphorus.

**Figure 5 plants-12-03870-f005:**
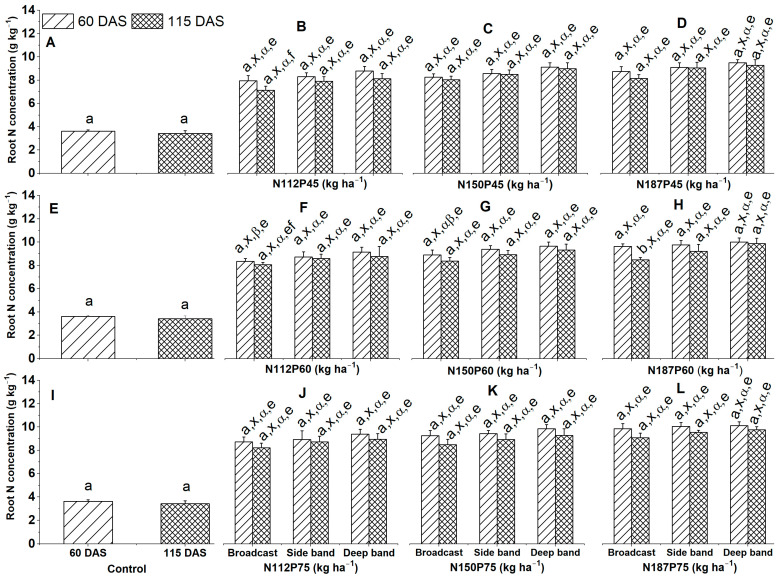
Effects of nitrogen fertilization rates and fertilizer placements on root N concentrations of maize in the VT (tasseling) stage at 60 days after sowing, and the R6 (physiological maturity or harvest) stage at 115 days after sowing. Figures (**A**,**E**,**I**) represent no-fertilization control treatment; (**B**–**D**) showed difference between different N fertilization rates and fertilizer placements for the same P45 fertilization rate; (**F**–**H**) between different N fertilization rates and fertilizer placements for the same P60 fertilization rate; (**J**–**L**) between different N fertilization rates and fertilizer placements for the same P75 fertilization. Since there were no significant differences between varieties (Xida-789 and Xida-211) and years (2018 and 2019), data are combined under three fertilizer placements for the control treatments and under the same N and P fertilizer for the same fertilizer placement. Data (means ± SE, *n* = 6) followed by different letters indicate significant differences between different growth days for the same N and P fertilization rate and same fertilizer placement or between 60 DAS and 115 DAS under the control (a, b), between different fertilizer placements for the same growth day and same N and P fertilization rate (x), between different N fertilization rates with the same P fertilization rate for the same fertilizer placement and plant growth day (α, β), and between different P fertilization rates with the same N fertilization rate for the same fertilizer placement and plant growth day (e, f, g) at *p* < 0.05. Abbreviations: DAS, days after sowing; N, nitrogen; P, phosphorus.

**Figure 6 plants-12-03870-f006:**
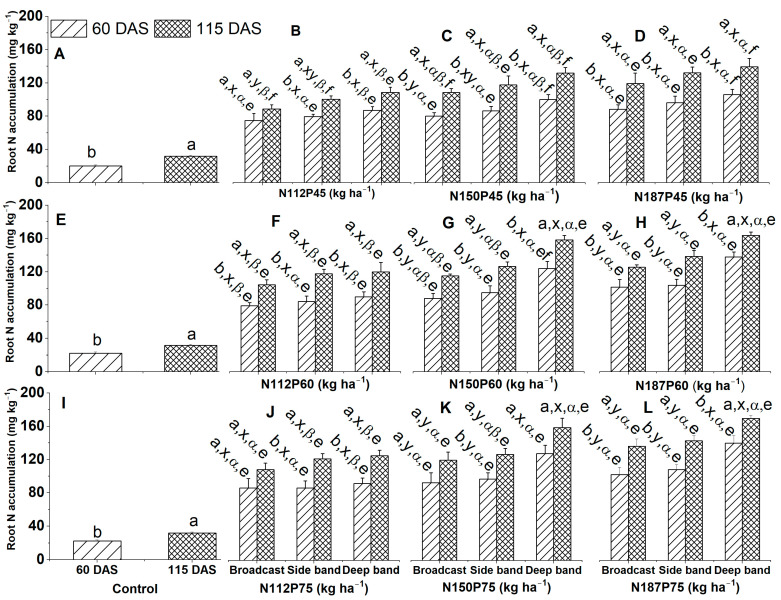
Effects of nitrogen fertilization rates and fertilizer placements on root N accumulations of maize in the VT (tasseling) stage at 60 days after sowing and the R6 (physiological maturity or harvest) stage at 115 days after sowing. Figures (**A**,**E**,**I**) represent no-fertilization control treatment; (**B**–**D**) showed difference between different N fertilization rates and fertilizer placements for the same P45 fertilization rate; (**F**–**H**) between different N fertilization rates and fertilizer placements for the same P60 fertilization rate; (**J**–**L**) between different N fertilization rates and fertilizer placements for the same P75 fertilization. Since there were no significant differences between varieties (Xida-789 and Xida-211) and years (2018 and 2019), data are combined under three fertilizer placements for the control treatments and under the same N and P fertilizer for the same fertilizer placement. Data (means ± SE, *n* = 6) followed by different letters indicate significant differences between different growth days for the same N and P fertilization rate and same fertilizer placement or between 60 DAS and 115 DAS under the control (a, b), between different fertilizer placements for the same growth day and same N and P fertilization rate (x, y), between different N fertilization rates with the same P fertilization rate for the same fertilizer placement and plant growth day (α, β), and between different P fertilization rates with the same N fertilization rate for the same fertilizer placement and plant growth day (e, f) at *p* < 0.05. Abbreviations: DAS, days after sowing; N, nitrogen; P, phosphorus.

**Figure 7 plants-12-03870-f007:**
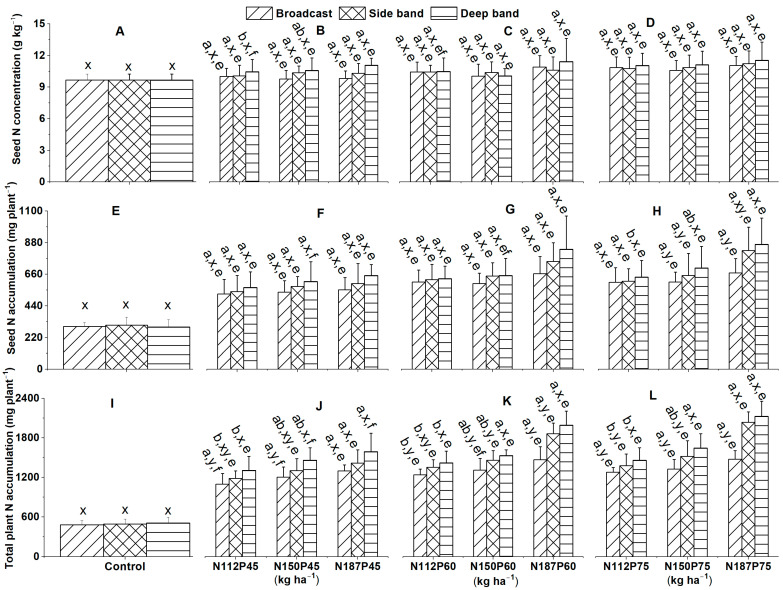
Effects of nitrogen fertilization rates and fertilizer placement on seed N concentrations (**A**–**D**), seed N accumulations (**E**–**H**), and total plant (leaf + stem + seed + root) N accumulations (**I**–**L**) of maize crop in the R6 (physiological maturity or harvest) stage at 115 days after sowing. Since there were no significant differences between varieties (Xida-789 and Xida-211) and years (2018 and 2019), data are combined under the three fertilizer placements for the control treatment and under the same N and P fertilization for the same fertilizer placement. Data (means ± SE, *n* = 6) followed by different letters indicate significant differences between different N fertilization rates for the same P fertilization and fertilizer placement (a, b), between different fertilizer placements for the same N and P fertilization rate (x, y), and between different P fertilization rates for the same N fertilization and fertilizer placement (e, f) at *p* < 0.05. Abbreviations: DAS, days after sowing; N, nitrogen; P, phosphorus.

**Figure 8 plants-12-03870-f008:**
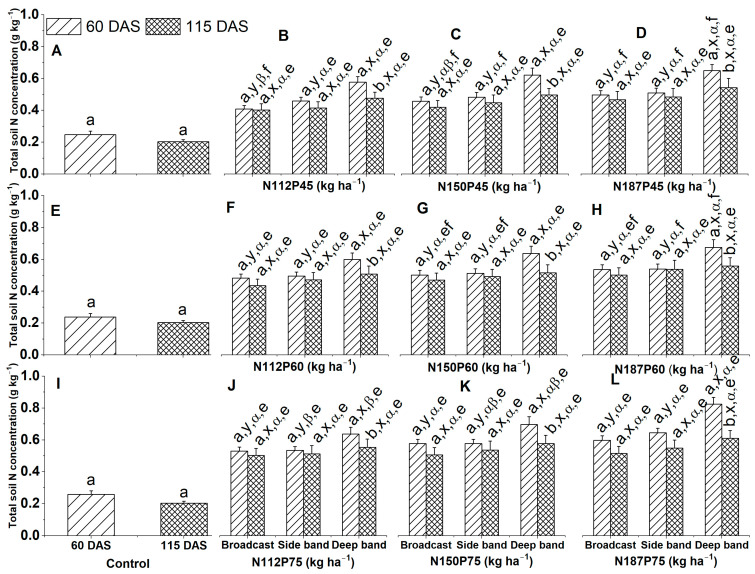
Effects of nitrogen fertilization rates and fertilizer placements on total soil N concentrations in the maize’s VT (tasseling) stage at 60 days after sowing, and the R6 (physiological maturity or harvest) stage at 115 days after sowing. Figures (**A**,**E**,**I**) represent no-fertilization control treatment; (**B**–**D**) showed difference between different N fertilization rates and fertilizer placements for the same P45 fertilization rate; (**F**–**H**) between different N fertilization rates and fertilizer placements for the same P60 fertilization rate; (**J**–**L**) between different N fertilization rates and fertilizer placements for the same P75 fertilization. Since there were no significant differences between varieties (Xida-789 and Xida-211) and years (2018 and 2019), data are combined under three fertilizer placements for the control treatments and under the same N and P fertilizer for the same fertilizer placement. Data (means ± SE, *n* = 6) followed by different letters indicate significant differences between different growth days for the same N and P fertilization rate and same fertilizer placement or between 60 DAS and 115 DAS under the control (a, b), between different fertilizer placements for the same growth day and same N and P fertilization rate (x, y), between different N fertilization rates with the same P fertilization rate for the same fertilizer placement and plant growth day (α, β), and between different P fertilization rates with the same N fertilization rate for the same fertilizer placement and plant growth day (e, f) at *p* < 0.05. Abbreviations: DAS, days after sowing; N, nitrogen; P, phosphorus.

**Figure 9 plants-12-03870-f009:**
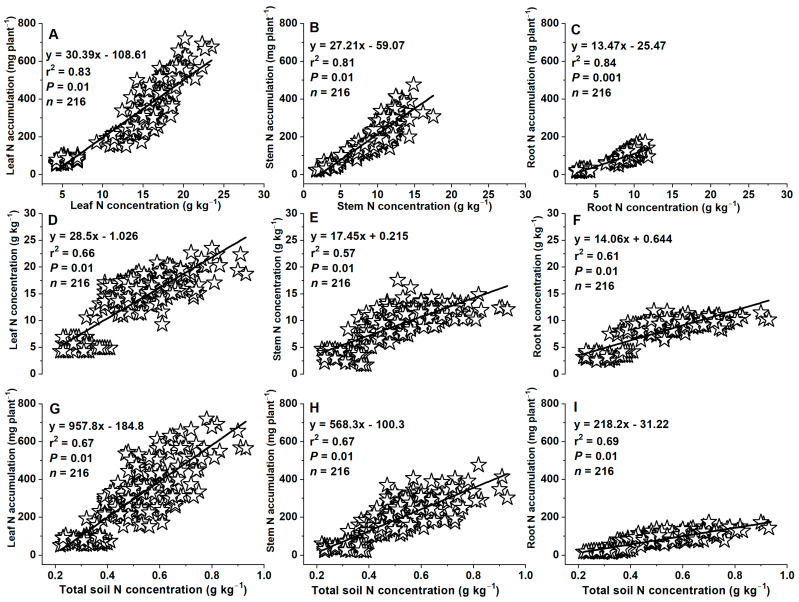
Relationships between tissue N accumulations and concentrations in leaf (**A**), stem (**B**), or root (**C**), between tissue N concentrations in leaf (**D**), stem (**E**), root (**F**), and total soil N concentrations, and between tissue N accumulations in leaf (**G**), stem (**H**) root (**I**), and total soil N concentrations in the VT (tasseling) growth stage at 60 days after sowing. Abbreviations: N, nitrogen; P, phosphorus.

**Figure 10 plants-12-03870-f010:**
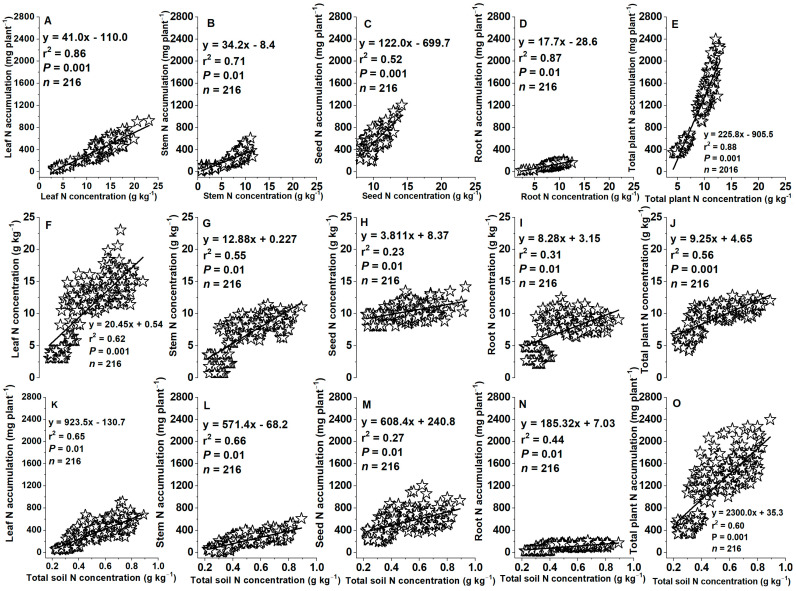
Relationships between tissue N accumulations and concentrations in leaf (**A**), stem (**B**), seed (**C**), root (**D**), total plant concentration (**E**), between tissue N concentrations in leaf (**F**), stem (**G**), seed (**H**), root (**I**), and total plant (leaf + stem + seed + root; (**J**) total soil N concentrations, and between tissue N accumulation in leaf (**K**), stem (**L**), seed (**M**), root (**N**), and total plant (leaf + stem + seed + root; (**O**) total soil N concentration in the R6 (physiological maturity or harvest at 115 days after sowing. Abbreviations: N, nitrogen; P, phosphorus.

**Figure 11 plants-12-03870-f011:**
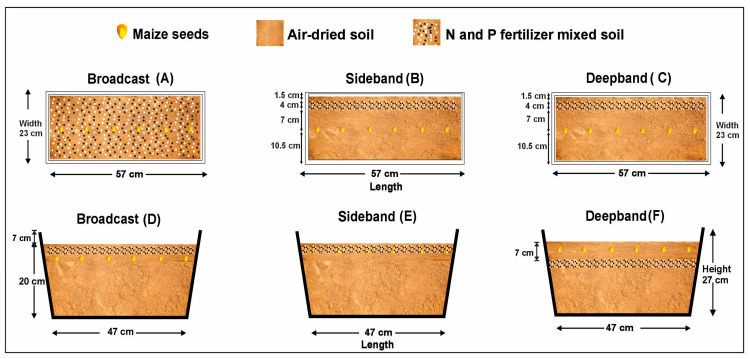
Experimental set-up of pots to grow maize plants under three contrasting fertilizer placement methodologies. Each pot (47 or 57 cm for the bottom or top length, 15 or 23 cm for the bottom or top width, and 27 cm of height) holds a total of 26 kg of air-dried soil plus an equal amount of nitrogen and phosphorus nutrition according to the N and P fertilization rates. (1) Broadcast: 5 kg N and P fertilizer mixed soils are evenly spread on the soil surface (**A**,**D**); (2) Side band: 0.5 kg N and P fertilizer mixed soils as a 4 cm narrow strip are buried on the soil surface with a 7 cm distance from or alongside the sowing line (**B**,**E**); and (3) Deep band: 0.5 kg N and P fertilizer mixed soils as a 4 cm narrow strip are buried at or below 7 cm soil depth with a 7 cm distance from or alongside the sowing line (**C**,**F**). Abbreviations: N, nitrogen; P, phosphorus.

**Table 1 plants-12-03870-t001:** Effects of nitrogen and phosphorus fertilization rates and fertilizer placement on nitrogen agronomy efficiency, nitrogen use efficiency, partial factor productivity of nitrogen of maize.

Nitrogen and Phosphorus Fertilization Rate	Fertilizer Placement	Nitrogen Agronomy Efficiency (kg Grain kg^−1^)	Nitrogen Use Efficiency (%)	Partial Factor Productivity of Nitrogen (kg Grain kg^−1^ N)
N112P45	Broadcast	14.2 ± 3.7 (a,x,e)	42.4 ± 5.3 (a,x,f)	35.8 ± 2.5 (a,x,e)
	Side band	14.9 ± 2.9 (a,x,e)	48.4 ± 4.6 (a,x,e)	37.5 ± 1.4 (a,x,e)
	Deep band	16.8 ± 3.2 (a,x,e)	52.5 ± 3.3 (a,x,f)	38.7 ± 2.1 (a,x,e)
N150P45	Broadcast	11.2 ± 1.4 (a,x,e)	36.1 ± 2.8 (a,x,f)	27.3 ± 1.5 (b,x,e)
	Side band	11.6 ± 2.2 (a,x,e)	41.3 ± 4.4 (a,x,e)	28.4 ± 1.1 (b,x,e)
	Deep band	13.1 ± 2.7 (a,x,f)	45.3 ± 4.6 (a,x,g)	29.4 ± 2.1 (b,x,f)
N187P45	Broadcast	9.5 ± 2.0 (a,x,e)	33.9 ± 3.9 (a,x,e)	22.4 ± 1.3 (b,x,e)
	Side band	10.2 ± 1.8 (a,x,e)	39.6 ± 2.8 (a,x,f)	23.7 ± 1.9 (c,x,e)
	Deep band	11.0 ± 1.8 (a,x,f)	43.5 ± 4.9 (a,x,f)	24.2 ± 0.9 (b,x,f)
N112P60	Broadcast	18.4 ± 3.0 (a,x,e)	51.4 ± 2.0 (a,x,ef)	40.0 ± 1.9 (a,x,e)
	Side band	18.4 ± 4.2 (a,x,e)	54.7 ± 6.5 (a,x,e)	41.0 ± 2.1 (a,x,e)
	Deep band	19.8 ± 3.0 (a,x,e)	65.2 ± 6.2 (a,x,ef)	41.7 ± 1.9 (a,x,e)
N150P60	Broadcast	14.5 ± 3.0 (a,x,e)	44.3 ± 2.3 (ab,y,ef)	30.6 ± 1.8 (b,y,e)
	Side band	15.3 ± 1.7 (a,x,e)	48.5 ± 4.0 (a,y,e)	32.2 ± 2.0 (b,xy,e)
	Deep band	19.8 ± 1.8 (a,x,e)	68.9 ± 2.9 (a,x,f)	36.2 ± 1.5 (b,x,e)
N187P60	Broadcast	12.0 ± 1.1 (a,y,e)	38.3 ± 3.5 (b,y,e)	24.9 ± 1.2 (c,y,e)
	Side band	13.0 ± 1.7 (a,xy,e)	41.9 ± 1.4 (a,y,ef)	26.6 ± 1.7 (b,xy,e)
	Deep band	16.5 ± 1.1 (a,x,e)	60.7 ± 3.1 (a,x,e)	29.6 ± 1.3 (c,x,e)
N112P75	Broadcast	15.6 ± 4.0 (a,x,e)	55.7 ± 2.8 (a,x,e)	38.2 ± 2.2 (a,x,e)
	Side band	17.4 ± 4.0 (a,x,e)	58.8 ± 5.4 (a,x,e)	39.9 ± 2.8 (a,x,e)
	Deep band	21.0 ± 4.0 (a,x,e)	69.1 ± 5.3 (b,x,e)	41.8 ± 2.5 (a,x,e)
N150P75	Broadcast	12.6 ± 4.4 (a,y,e)	46.6 ± 3.3 (b,y,e)	29.5 ± 2.0 (b,y,e)
	Side band	13.7 ± 3.0 (a,y,e)	53.8 ± 3.6 (a,y,e)	30.5 ± 1.9 (b,y,e)
	Deep band	22.0 ± 1.2 (a,x,e)	80.3 ± 2.4 (a,x,e)	37.6 ± 1.9 (a,x,e)
N187P75	Broadcast	10.4 ± 1.7 (a,y,e)	40.8 ± 2.8 (b,y,e)	23.9 ± 1.3 (b,y,e)
	Side band	12.4 ± 0.9 (a,y,e)	48.3 ± 3.2 (a,y,e)	25.9 ± 1.6 (b,y,e)
	Deep band	18.6 ± 1.8 (a,x,e)	68.3 ± 2.2 (b,x,e)	31.0 ± 1.8 (b,x,e)

Since there were no significant differences between varieties (Xida-789 and Xida-211) and years (2018 and 2019), data are combined under the three fertilizer placements and under the same N and P fertilization for the same fertilizer placement. Data (means ± SE, *n* = 6) followed by different letters indicate significant differences between different N fertilization rates for the same P fertilization and fertilizer placement (a, b, c), between different fertilizer placements for the same N and P fertilization rate (x, y), and between different P fertilization rates for the same N fertilization and fertilizer placement (e, f, g) at *p* < 0.05. Abbreviations: N, nitrogen; P, phosphorus.

**Table 2 plants-12-03870-t002:** Monthly temperature, rainfall, and humidity during the maize growing season in 2018 and 2019.

Months	Precipitation (mm)	Temperature (°C)	Humidity (%)	Precipitation (mm)	Temperature (°C)	Humidity (%)
		2018			2019	
May	201.8	22.5	77.7	203.3	20.4	83.1
June	96.1	25.3	77.3	242.4	24.2	83.4
July	114.7	30.4	66.4	176.9	26.9	79.5
August	120.3	29.2	68.9	53.1	29.5	66.3
September Means ± SD	183.2 143.2 ± 46.3	22.7 26.0 ± 3.7	83.6 74.8 ± 7.0	109.7 157.1 ± 75.6	23.8 25.0 ± 3.4	77.3 77.9 ± 7.0

## Data Availability

Data are contained within the article.

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
