# Peer review of "Variations in Nitrogen Accumulation and Use Efficiency in Maize Differentiate with Nitrogen and Phosphorus Rates and Contrasting Fertilizer Placement Methodologies"

_plants, 2023, doi:10.3390/plants12223870_

Round 1
Reviewer 1 Report
Comments and Suggestions for Authors
Report about:
Variation in nitrogen accumulation and use efficiency in maize differentiate with nitrogen and phosphorus rates and contrasting fertilizer placement methodologies
2617966
The aim of the paper was to identify the optimum NP fertilization rate; to know a practical fertilizer placement methodology; and then to determine a positive correlation between an appropriate NP rate with proper placement and tissue N accumulations to increase maize production. The generated results from the current work will contribute to field practices in exploring effective fertilization rate and methodology for small-holder farmers to increase crop production.
This is a very well written review paper, and I enjoyed reading it. This is a very interesting article touching a very hot and applied point. It was a pleasure to read. The figures are excellent, and the tables are also very good but need to be improved as below. This paper contains nicely designed tables and figures. I consider the topic of this paper original and relevant to the field and suitable for the journal. The paper addressed a specific gap in the field. The conclusions are consistent with the evidence and arguments presented and the authors addressed all the main questions posed.
My comments to improve the review:
Please make all tables self-explanatory and do not use any abbreviation in the table footnote or in the table legend. Please make all figures self-explanatory and do not use any abbreviation in the figure footnote or in the figure legend.
All references are appropriate, but I suggest adding more 2022 and 2023 references.
Please change the title. The title is not correct and not easy to follow.
Some other comments.
Keywords
Please arrange all key words in alphabetical order.
Please add new references 2023 please.
Please make all tables self-explanatory.
Please make all tables self-explanatory.
In addition, many scientific names in the references must be written in italic format
In order to improve the quality of the paper update the reference list by adding 2022 and 2023 references.
Please follow the instructions to authors on how they write the reference in the list. For references about textbooks, please add the page numbers of the textbook. Also please add the city of the publisher.
This paper can be accepted with minor revision, but I must revise the paper once more to ensure that all my comments were incorporated.
Comments on the Quality of English Language
English is OK. Minor editing is needed.
Author Response
Comments from Reviewer 1.
Comment 1. Please make all tables self-explanatory and do not use any abbreviation in the table footnote or in the table legend. Please make all figures self-explanatory and do not use any abbreviation in the figure footnote or in the figure legend.
Response: Thanks the necessary changes have been now made to all the legends of figures and tables. At the same time, all relevant abbreviations have been therefore added to reflect their meanings in the figure and tables (Page number 4 – 20).
Comment 2. All references are appropriate, but I suggest adding more 2022 and 2023 references.
Response: Thanks and a total of eight new references between 2022 and 2023 have been now added to the Introduction, Discussion and Reference sections of the revised manuscript.
- Fathi 2022 (see line 46, 1506 - 1507);
- Li et al. 2022 (see lines 54-55, 1196-1199);
- Battisti et al. 2022 (see lines 85, 1225, 1233);
- Hou et al. 2023 (see lines 1119-1122);
- Wu et al. 2022 (see line 1150-1151);
- Wang et al. 2023 (see line 1150-1151);
- BarÅ‚óg et al. 2022 (see line 1249-1250);
- Pandit et al. 2022 (see line 1252-1254).
Comment 3. Please change the title. The title is not correct and not easy to follow.
Response: Thanks, we are sorry for the grammar error in the title, and we have now revised title as “Variations in nitrogen accumulation and use efficiency in maize differentiate with nitrogen and phosphorus rates and contrasting fertilizer placement methodologies”.
It would be greatly appreciated if you would suggest a suitable manuscript title.
Comment 4. Please arrange all key words in alphabetical order.
Response: Done, thanks.
Comment 5. Many scientific names in the references must be written in italic format.
Response: Done, thanks.
Comment 6. Please follow the instructions to authors on how they write the reference in the list. For references about textbooks, please add the page numbers of the textbook. Also please add the city of the publisher.
Response: Done, thanks.
Reviewer 2 Report
Comments and Suggestions for Authors
The experiment addresses important issues concerning localized/row fertilisation of nitrogen and phosphorus in maize cultivation. This issue has been the subject of research on the effectiveness of fertilization of this demanding plant for years. Due to the maize extensive root system – both vertically and horizontally – studies regarding the use of ingredients should be carried out in field conditions, as the authors rightly noted in the summary. The intention of the authors was to fully control the growth of the plant (roots and above-ground biomass), but in the case of maize such studies are of little value, because the conditions in the greenhouse (more precisely, in pots) differ significantly from the actual conditions occurring in the field. As early as in 1926, Weaver (Root development of field crops) determined the depth of the maize root system on day 36 at 30 cm, and after 8 weeks of vegetation the roots penetrated the soil to a depth of 90 cm. This means that in the later developmental phases, which the authors took samples in, the obtained results do not correspond to the natural exploration of the soil by plants. That, in turn, means that the result must be flawed with an error.
Author Response
Comments from Reviewer 2.
The experiment addresses important issues concerning localized/row fertilization of nitrogen and phosphorus in maize cultivation. This issue has been the subject of research on the effectiveness of fertilization of this demanding plant for years. Due to the maize extensive root system – both vertically and horizontally – studies regarding the use of ingredients should be carried out in field conditions, as the authors rightly noted in the summary. The intention of the authors was to fully control the growth of the plant (roots and above-ground biomass), but in the case of maize such studies are of little value, because the conditions in the greenhouse (more precisely, in pots) differ significantly from the actual conditions occurring in the field. As early as in 1926, Weaver (Root development of field crops) determined the depth of the maize root system on day 36 at 30 cm, and after 8 weeks of vegetation the roots penetrated the soil to a depth of 90 cm. This means that in the later developmental phases, which the authors took samples in, the obtained results do not correspond to the natural exploration of the soil by plants. That, in turn, means that the result must be flawed with an error.
Response: Thanks for your concerns, and the reference of “Weaver, J.E. 1926. Root habits of corn or maize. In Root Development of Field Crops, 1st ed.; McGraw-Hill Book Company, Inc., New York, USA”.
Yes, indeed, the growth of roots will be heavily restricted by pot volumes, although root penetration is comparatively easier due to a crushed and mixed soil medium before being potted. However, compared to a field experiment under varied or multiple abiotic and biotic variables, a potted experiment under a greenhouse environment would be easier to manipulate the dynamics of temperature, water, and nutrient supplement, with comparatively less laborious and relatively precise, although cannot reflect those in a field habitat. Perhaps, a combination of potted with field experiments could provide more useful practice guidance to improve crop plantations.
As a result, the following writings have been added to the Conclusion section: “Nevertheless, since soil and plant growth conditions in greenhouses are largely different from those in the field, usefully information based on pot experiments, must be cautiously analogized or referred to field practice, and future field experiments are therefore timely required to further evaluate the effect of N and P fertilization under placement methodology on promoting crop production”.
Reviewer 3 Report
Comments and Suggestions for Authors
The over doses of chemical fertilizers along with incorrect methods of application not only contribute to lose of fertilization without significant improvement in crop yield but also increase the chances of air pollution and waterbody eutrophication. Therefore, optimization of chemical fertilization rate and methods of application for each crop separately is vital. In this article, Sharifi et al. had an attempt to explore the effect of N and P fertilization and their placement methods on maize crop. This study presented two years of experimental data covering the different plant tissues (roots, leave, stem and seeds) and different growth stages, which is always desirable. The experimental material and methods are presented in detail and results are well illustrated and interpreted and are discussed in detail. Also, the research content of this article is well in line with the journal scope. Therefore, I recommend it for publication in current form.
Author Response
Comments from Reviewer 3.
The over doses of chemical fertilizers along with incorrect methods of application not only contribute to lose of fertilization without significant improvement in crop yield but also increase the chances of air pollution and waterbody eutrophication. Therefore, optimization of chemical fertilization rate and methods of application for each crop separately is vital. In this article, Sharifi et al. had an attempt to explore the effect of N and P fertilization and their placement methods on maize crop. This study presented two years of experimental data covering the different plant tissues (roots, leave, stem and seeds) and different growth stages, which is always desirable. The experimental material and methods are presented in detail and results are well illustrated and interpreted and are discussed in detail. Also, the research content of this article is well in line with the journal scope. Therefore, I recommend it for publication in current form.
Response: We are very grateful for your comment and recommendation.
Reviewer 4 Report
Comments and Suggestions for Authors
Dear Authors
The manuscript " Variation in nitrogen accumulation and use efficiency in maize differentiate with nitrogen and phosphorus rates and contrasting fertilizer placement methodologies" is quite carefully written, and deals with a very important topic nitrogen efficiency. The use of nitrogen and its application in soil or understanding and optimizing nitrogen use efficiency in maize production is essential for sustainable agriculture and food security.
Introduction is well organized with well-established hypotheses and with sufficient explanation of the nitrogen effects maize yeald.
Measurements, laboratory and statistical analyses in MATERIAL AND METHODS section are quite well done, although it will be good to explain how you calculated NAE, NUE and PFPN. Also, it will be good to better describe - explain the type of soil in physical and chemical terms. For better understanding the whole picture it will be also good to conduct Principal components analysis PCA
Results are concisely present with well-designed and explained tables and figures. In the discussion section all result is well explain.
Summary
I recommend the manuscript for publication in Plants after minor revision.
Author Response
Comments from Reviewer 4.
Comment 1. It will be good to explain how you calculated NAE, NUE and PFPN.
Response: Thank you for your concern. The detailed calculations of NAE, NUE and PFPN have been now added in the 4.5 subsection of the Materials and Methods section of the revised manuscript (see lines 1406 – 1421).
Comment 2. It will be good to better describe - explain the type of soil in physical and chemical terms.
Response: Thanks, and the relevant information on the formation of the used soil and its basic soil physiochemical properties have been now added in the Section 4.1 and 4.2. (see lines 1337– 1330 and 1353 – 1354), respectively.
Comment 3. For better understanding the whole picture, it will be also good to conduct Principle components analysis PCA.
Response: Thanks for your valuable suggestion. However, a PCA has not been run between plant and soil parameters considering that (1) the Figure 9 and 10 have already shown the relationships between tissue nitrogen (N) accumulations and tissue N concentrations, between tissue N concentrations or accumulations in leaf, stem, root, seed (i.e. grain yield), or total plant (leaf + stem + seed + root; O) and total soil N concentration in both the tasseling (60-days-old) and physiological maturity (115-days-old) stages; and (2) the generated results from this environment-controlled pot experiment cannot directly apply to the real filed, where a PCA can establish better meaningful relationships between biotic and abiotic variables for explaining the main component to affect plant growth).
Round 2
Reviewer 2 Report
Comments and Suggestions for Authors
Adding one sentence in the "Conclusions" chapter does not change anything in my assessment of the research methodology. As I have already mentioned, the obtained research results are subject to error.